



# The Climatology of Australian Aerosol

Ross M. Mitchell[1], Bruce W. Forgan[2], and Susan K. Campbell[1]

[1]CSIRO Oceans and Atmosphere, Yarralumla, GPO Box 1700, Canberra, ACT, Australia
[2]Australian Bureau of Meteorology, PO Box 1289K, Melbourne, Australia

*Correspondence to:* R. M. Mitchell (Ross.Mitchell@csiro.au)

**Abstract.** Airborne particles or aerosols have long been recognized for their major contribution to uncertainty in climate change. In addition, aerosol amounts must be known for accurate atmospheric correction of remotely sensed images, and are required to accurately gauge the available solar resource. However, despite great advances in surface networks and satellite retrievals over recent years, long-term continental-scale aerosol data sets are lacking. Here we present an aerosol assessment
over Australia based on combined sun photometer measurements from the Bureau of Meteorology Radiation Network and CSIRO/AeroSpan. The measurements are continental in coverage, comprising 22 stations, and generally decadal in time-scale, totalling 207 station-years. Spectral decomposition shows that the time series can be represented as a weighted sum of sinusoids with periods of 12, 6 and 4 months, corresponding to the annual cycle and its second and third harmonics. Their relative amplitudes and phase relationships leads to sawtooth-like waveforms sharply rising to an austral spring peak, with a slower
decline often including a secondary peak during the summer. The amplitude and phase of these periodic components show significant regional change across the continent. Fits based on this harmonic analysis are used to separate the periodic and episodic components of the aerosol time series. Classification of the aerosol types is undertaken based on (a) the spectral variation of the optical depth expressed in the Ångström exponent, (b) the Fourier decomposition, and (c) the ratio of episodic to periodic variation in aerosol optical depth. It is shown that Australian aerosol can be broadly grouped into three classes:
Temperate, Arid, and Tropical. The Temperate class is characterised by a small amplitude periodic component, with an increasing episodic component toward the fire-prone Eucalypt forests of the south-east. Arid zone aerosol has a larger periodic component, with pronounced twin spring-summer peaks, and an increasing episodic component towards active dust source regions. Tropical aerosol is characterised by a very large periodic component due to seasonal biomass burning in the savanna belt, with significant interannual variability due to variation in the strength of the monsoon and its effect on the fuel source.
Statistically significant decadal trends are found at 4 of the 22 stations. Despite the apparently small associated declining trends in mid-visible aerosol optical depth of between 0.001 to 0.002 per year, these trends are much larger than those projected to occur due to declining emissions of anthropogenic aerosols from the northern hemisphere. There is remarkable long-range coherence in the aerosol cycle across the continent, suggesting broadly similar source characteristics, including a possible role for inter-continental transport of biomass burning aerosol.



# 1  Introduction

Suspended atmospheric particles or aerosols are extensively studied because of their multiple environmental roles, and because - unlike well-mixed gases - their concentration and properties vary widely over space and time. Aerosols adversely affect human health (particularly in urban areas), and exert a further range of impacts due to their modulation of electromagnetic radiation

within the atmosphere. Scattering and absorption by aerosol remains a major source of uncertainty in climate projections (Boucher et al., 2013), while satellite images of the Earth's surface require correction for aerosol effects. In addition, aerosol data are also required for the accurate assessment and forecasting of available solar energy (e.g., Perry and Troccoli, 2015).

Major enhancements in surface networks and satellite retrievals over the last 15 years have radically boosted knowledge of global aerosol. In particular, surface networks including AERONET (Holben et al., 1998), SKYNET (Nakajima et al., 2007;

Hashimoto et al., 2012) and GAW/PFR (WMO, 2005) have provided regional aerosol climatologies and supported process studies (e.g., Eck et al., 2008), while several major satellite programs have championed the characterization of aerosols from orbit. Of note in this respect are NASA's flagship EOS A-train instruments MODIS (Levy et al., 2010), MISR (Martonchik et al., 2009) and CALIPSO (Young et al., 2013). In addition, the ATSR series of instruments operated by ESA has made a significant contribution to satellite aerosol measurement from space, particularly the AATSR instrument launched on the

ENVISAT platform in March 2002 (Grey and North, 2009; Qin et al., 2015). However, retrieval of aerosol from satellites remains challenging (Mishchenko et al., 2009), and requires ongoing availability of validation data from surface networks. In addition to these more recent sensors, the 30-year record available from the NOAA/AVHRR series is proving valuable, as shown by the recent examination of the aerosol indirect effect of maritime aerosol on cloud properties (Zhao et al., 2016).

Since aerosol emission is highly correlated with human population and land mass, there has been a striking northern-

hemisphere bias in aerosol characterisation efforts to date. However, there is a growing need for improved characterisation of southern hemisphere aerosol. In particular, with lower levels of anthropogenic aerosol (AA), the southern hemisphere is on average less subject to 'masking' of the greenhouse warming than the northern hemisphere (Rotstayn et al., 2013). Consequently, low AA regions including Australia experience the greenhouse gas component of warming with less moderation from aerosol radiative forcing, to a degree dependent on improved knowledge of Australian natural aerosol. Rotstayn et al. (2013)

sought to quantify the unmasking implied by projected future decreases in AA, and their findings will be further considered below. Furthermore, recently Stephens et al. (2015) showed that the contribution of each hemisphere to the planetary albedo averaged over the last 13 years has been in near-perfect balance, with the increased reflection from northern hemisphere land exactly balanced by increased atmospheric scattering in the southern hemisphere. While the full ramifications of this finding are yet to be fully explored, there is a pressing need to better characterise southern hemisphere aerosol emissions.

Australia is the largest dust source in the southern hemisphere (Tanaka and Chiba, 2006) and is a significant source of smoke aerosol from savanna burning (Mitchell et al., 2013). Hence it is clear that Australia's aerosol emissions make up a significant fraction of the southern hemisphere total. Since aerosol contributes to the planetary albedo both directly, and indirectly through the modulation of cloud properties, the need for improved continental-scale characterisation of Australian aerosol is clear.



Studies of Australian aerosol to date have often been restricted in time and space. While coastal maritime aerosol has been extensively studied (Gras and Ayers, 1983; Gras, 1991), studies of the continental aerosol are more restricted. Thus Scott et al. (1992) characterised the aerosol cycle caused by seasonal biomass burning at Broome in the tropical north west of Australia in the late 1980s, with more extended analysis of regional tropical aerosol carried out by Mitchell et al. (2013); see also references cited therein. Studies of the smoke from Eucalypt fires in southern temperate forests are few in number, but include Mitchell
et al. (2006) and Radhi et al. (2006).

Australian dust emission has been well-studied from a geomorphological perspective (e.g., Leys et al., 2008; Bullard et al., 2008) but less so via sun photometry, where the measurand – column aerosol optical depth – is the required input for climate and remote sensing studies. However, Mitchell et al. (2010) analysed sun photometer and nephelometer measurements from Tinga Tingana in the Australian arid zone during the Millennium drought (2002-2010) and found an approximate doubling in
both column and near-surface aerosol during the austral summer over the duration of the drought. Feedbacks related to dust aerosol were modelled by Rotstayn et al. (2012), who found that inclusion of interactive dust in the model amplifies the impact of the ENSO cycle on Australian climate.

While satellite retrieval of aerosol over Australia is promising in terms of the extensive areal and temporal coverage available over a sparsely-populated continent, the dense-dark vegetation (DDV) assumption inherent in the MODIS aerosol retrieval
algorithm does not apply well over most of Australia (Levy et al., 2010; Gillingham et al., 2012). Problems in other sensors and retrieval methods are discussed by Qin et al. (2015), who show that the dual-view scan geometry of the AATSR instrument offers particular advantages under Australian conditions. However, the ongoing need for validation data from surface networks is universal.

Trends in aerosol loading over the period 1995–2009 were recently compiled by Yoon et al. (2016) at 15 AERONET stations.
They showed that fine-mode anthropogenic aerosol is decreasing over those countries with strict environmental regulations, and vice-versa. Coarse-mode dominant aerosol depends strongly on meteorological conditions subject to climate change, with stations close to regions of increasing desertification showing a significant increase in coarse-mode dominant aerosol.

In this paper we present a continent-wide climatology of Australian aerosol obtained at 22 stations, spanning all mainland states, and comprising a total of 207 station-years. The stations are operated by two agencies, the Australian Bureau of Mete-
orology, and the Commonwealth Scientific and Industrial Research Organization (CSIRO). The CSIRO stations are affiliated with NASA's worldwide AERONET sun photometer system, as further described below. Section 2 describes the two measurement systems and the data processing applied to each. The result of this step is a time series of multi-spectral monthly mean aerosol optical depth. Section 3 describes the common analysis applied to these time series. Section 4 presents a discussion of the results, with conclusions given in Section 5. The Appendix presents the monthly aerosol climatology at each of the 22
stations in tabular form.





## 2   Observations

This study is based on aerosol measurements obtained at 22 stations across the Australian continent between July 1997 and June 2015. The details of the 22 stations are given in Table 1. The columns headed 'First' and 'Last' denote the extremities of the data record for a given station, but not its continuity. The right-hand column labelled 'Months' gives the number of months for which sufficient observations were made to generate a valid monthly mean. A given month is deemed to have a valid monthly mean based on (a) the number of days in the month on which there were sufficient ($\geq$8) point measurements of aerosol optical depth to report a daily mean (NumDayOK), and the total number of point aerosol optical depth measurements in the month (NumObsMonth). The criterion applied in this study was *either* (NumDayOK$\geq$8 & NumObsOK>300) *or* (NumDayOK$\geq$6 & NumObsOK>600). The extent of spatial and temporal coverage is displayed in Figure 1, where the radius of the filled circles indicate the number of months reported by each station. The stations are listed in latitudinal order, starting in the tropics.

The measurements were obtained from sun photometers operated by the Bureau of Meteorology (abbreviated as 'Bureau' in Table 1 and later in the text) and CSIRO. The Bureau of Meteorology uses Middleton Solar SPO2 4-channel radiometers, with initially one and currently two radiometers at each site including nominal wavelengths of 368, 412, 500, 610, 778, 812 and 868 nm. The 500 nm channel is common to all radiometers and aids in cross calibration. The 610 nm channel is used with the 500 and 778 nm to provide an initial estimate of total column ozone, and the 812 nm channel is used to determine column water vapour. The FWHM for the filters is typically 10 nm, and an additional infrared blocking filter is used for the 368 nm channel to eliminate any infrared transmission given the low spectral irradiance at 368 nm.

The sun photometers operated by CSIRO are Cimel CE-318. The network is collectively known as AeroSpan (Aerosol characterisation via Sun Photometry: Australian Network) which forms the Australian component of AERONET, a worldwide sun photometer network operated by NASA (Holben et al., 1998), with contributing sub-networks from several countries outside the USA. The spectral channels used on the Cimel instrument have become standardized, and now include 340, 380, 440, 500, 670, 870, 1020 and 1640 nm, with a further channel at 936 nm yielding information on water vapour absorption. The channels between 340 and 1020 nm employ a silicon detector, while the 1640 nm channel uses an Indium gallium arsenide (InGaAs) detector. FWHM was likewise 10 nm for all channels.

Calibration of sun photometers refers to methods used to determine the exoatmospheric response of the instrument to the solar irradiance in a given spectral channel. This is a crucial procedure since once complete, the ratio between the actual instrument response and the exoatmospheric response gives the total atmospheric extinction, from which the aerosol extinction can be derived given knowledge of the gaseous absorption and scattering.

The use of different radiometers, wavelengths, and sampling strategies by the two networks introduces traceability issues when amalgamating the data series. In principle, the same calibration methodologies are used for in situ calibration as described in a detailed inter-calibration study of the two systems by Mitchell and Forgan (2003) at the Alice Springs station in 2000. The central result of the study was that the aerosol optical depths agreed to within 0.007 at the 95% confidence level for the two common wavelengths 500 nm and 868 nm. This well within the WMO (2005) requirement, and Bureau participation in the four WMO Filter Radiometer Comparisons held at the World Radiation Center in Davos, Switzerland, every 5 years begin-





ning in 2000, confirm the Bureau of Meteorology methodologies produce aerosol optical depth within the WMO uncertainty requirement.

Both networks utilise a combination of Langley methods to find a reference wavelength calibration and then apply the General Method developed by Forgan (1994) to calibrate the other wavelength channels (excluding the channels affected by water vapour). The General Method assumes the stability of the relative aerosol size distribution - but not the magnitude of the optical depth - throughout a morning or afternoon, and uses the reference wavelength aerosol optical thickness as the predictor for the least squares solution of the wavelength calibration value at the top of the atmosphere. As a Langley-type method involves an interpolation to the top of the atmosphere in real atmospheric conditions the selection of points not impacted by water or ice particles is key to the uncertainty of the top of atmosphere value. The two networks use different sampling strategies and methods to select points for the calibration process. The operational measurements are used to provide the in-situ calibrations.

The Bureau of Meteorology radiometers are co-located at solar and terrestrial irradiance stations that utilise the WMO Global Climate Observing System (or Baseline Surface Radiation Network) solar monitoring methodologies (Ohmura et al., 1998). The solar trackers employ active tracking sensors that mimic a 868 nm radiometer but with a wide field of view, and sample at 1 Hz. The standard deviation of the active tracking sensor is recorded every minute with the 0.12 s samples from the SPO2 radiometers. If the active sensor was not available the standard deviation of the direct irradiance from a pyrheliometer is used as an equivalent indicator of variability in the minute prior to the SPO2 measurement. The first pass of the selection process of data to use for SPO2 calibration is to eliminate the near zero SPO2 signals that are less than 5 times the measurement resolution (typically 0.005 V), and those periods when the standard deviation of the active sensor signal is more than 0.005 V. As the 868 nm active sensor signals in clear sun in typical Australian conditions is within 25% of the solar noon value for aerosol air masses less than 6, use of a fixed standard deviation is sufficient. The remaining data for each wavelength are then subject to the Harrison and Michalsky (1994) algorithm. If sufficient data remain, namely at least 30 points for aerosol air masses less than 6 and greater than 2, the calibration process is applied and the result recorded.

The AeroSpan measurements are based on a 'triplet' of the routine measurements obtained 30 s apart, spanning a minute. The standard deviation among the three measurements comprising the triplet is used for analogous filtering, as described in more detail by Mitchell and Forgan (2003).

For both networks, determinations of the natural logarithm of the top of atmosphere signal $(\ln V_0)$ at 1 AU earth-sun distance are compiled during the deployment period, which may range from months to over a year. The time series of the $\ln V_0$ is then used to derive the time dependence of the $\ln V_0$ through linear interpolation using a single slope where possible, or a series of linear functions if the calibration is changing in a non-linear manner. For some wavelengths of some radiometers the non-linear changes were rapid and difficult to model; these periods have been eliminated from the data analysis. The calibration functions for each wavelength providing a top of atmosphere value for each day are used in the derivation of optical depth for the entire measurement data set, not just the points selected for the calibration process.

Determination of aerosol optical depth requires subtraction of all relevant molecular extinction. For the present 'aerosol' channels (i.e., not including the 812/936 nm water vapour channels), these are nitrogen dioxide and ozone. The correction for





$NO_2$ is small (typically $<0.002$ at 380-440 nm and $<0.001$ at 500 nm (Mitchell and Forgan, 2003) and was excluded from the present analysis; however ozone absorption is more significant. The total column ozone determined from the Total Ozone Monitoring Satellite (TOMS) measurements was used in this analysis. In particular, TOMS-EP (Earth Probe) was used from 1996 to September 2004, while TOMS-OMI (Ozone Monitoring Instrument) was used from October 2004 to present. Currently, OMI Satellite Level 3e daily averaged ozone data gridded at $0.25°$ x $0.25°$ are obtained from the NASA Goddard Earth Sciences Data and Information Services Center Website. The ozone value used was based on the satellite data point nearest to the station coordinates.

For the elimination of cloud-affected aerosol optical depth measurements the two networks use different methods. In AeroSpan the filtering is similar to that described by Smirnov et al. (2000) used by AERONET. This first evaluates the short-term temporal variation based on triplets, then applies a further test based on the daily standard deviation of the aerosol optical depth. For the Bureau data the method of Alexandrov et al. (2004) is used after the elimination of signals less than 5 times the measurement resolution. This applies a moving temporal window of 15 minutes in span across a time series examining the variance in the initial estimate of the aerosol component of the optical depth.

## 2.1 Calculation of Ångström exponent

The Ångström exponent $\alpha$ captures the spectral slope of the aerosol optical depth, and is defined as

$$\alpha = -\frac{\ln(\tau_2/\tau_1)}{\ln(\lambda_2/\lambda_1)} \tag{1}$$

where $\tau$ is the spectral aerosol optical depth, and $\alpha$ is defined across the wavelength range ($\lambda_1$-$\lambda_2$). For the AeroSpan stations, the Ångström exponent was defined using the wavelength pair (440-870 nm), in part because these channels have always been present on the CE-318 instruments. For the Bureau stations, the Ångström exponents were calculated using the 500 nm wavelength and the 868 or 870 nm channel when available. For the Bureau data sets prior to 2009, 868 nm was only available from the active sensor, and led to an underestimate of the true optical depth due to the large opening angle. A comparison of the Bureau Ångström exponents using 500-778 and 500-868 was possible when the 868 nm channel was added in the last decade of measurements; no statistically significant difference was found. As a result, in the period before 2009, when the 868 nm was only available from the active sensor at Bureau sites, the 500-778 nm pair was used to calculate the Ångström exponent statistics reported below. A similar rationale supports the comparability of the AeroSpan and Bureau Ångström exponents, even though the basis wavelength pairs differ.

For the majority of the 22 stations, the annual mean aerosol optical depth at the 500 nm wavelength is less than 0.06, and at 868 nm less than 0.03. Hence even with the low uncertainties ($\sim$0.007) for the individual aerosol optical depth measurements, the uncertainties in the individual Ångström exponent measurements can be relatively large ($>$0.2). The monthly statistics of Ångström slopes based on individual values was compared to the monthly mean wavelength pair gradients derived from the General Method calibrations. The differences in monthly means were typically less than 0.1 except for those months where the mean aerosol optical depth at 500 nm was less than 0.020. The comparison of the Ångström exponent methods of measurement and subsequent statistics is the subject of future work.



## 3  Analysis

In this section we introduce the data sets, then define the model fitting procedure used as a basis for the analysis. This allows separation of the observed aerosol times series into a periodic component, with the residual interpreted as an episodic component. This is used as a basis for classification of the aerosol across the continent in terms of (a) the periodic amplitudes found for the Ångström exponent (a measure of aerosol size) and aerosol optical depth; (b) the harmonic content of the periodic component of the aerosol optical depth; and (c) the balance between the episodic and periodic components of the aerosol optical depth.

### 3.1  Monthly time series

Since the object of this work is to define the monthly climatology of Australian aerosol, aerosol time series were preprocessed into monthly means for subsequent analysis. The analysis involved (a) model fitting (as described below) and (b) spectral decomposition using a Fast Fourier Transform (FFT). Due to instrument malfunction and occasional removal for calibration, these time series contain infrequent gaps. The model fitting was carried out using all available data, simply ignoring periods when no data were available. However, the FFT required a complete time series, so interpolation for missing data was required. This was carried out using a combination of spline fitting, and fits based on Kalman filtering, as provided in the **R** package. Testing showed that the interpolation had no significant effect on the Fourier analysis, which was employed only to assess the spectral content of the aerosol time series.

Examples of the time series are shown in figures 2, 3 and 4. The left-hand panels show the monthly time series (black), together with the (annually repeated) climatology (red), and the fit (green, see below). Interpolated periods are shown in blue. The right-hand panels show the power spectrum obtained from the FFT. From the latter, the dominance of the annual cycle is clear, with significant contributions at the 6 and 4-month periods, varying between sites. This general pattern was evident at all sites. The increasing complexity of the power spectrum at Wagga Wagga (Figure 4) arises due to the high incidence of episodic smoke plumes from forest fires in the south eastern Eucalypt forests. Other stations - not shown here - exhibit similar characteristics, supporting the analysis outlined here.

### 3.2  Model fitting

The equation describing the model fit is as follows:

$$f(t) = a_0 + a_1 t + \sum_{j=1}^{J} b_j \cos[2\pi\omega_j(t - t_j^*)] \tag{2}$$

where $a_0$ is the time-invariant offset of the fit to the (dimensionless) aerosol optical depth time series, and $a_1$ is the linear coefficient with units of inverse time. The following term represents the sum of $J$ sinusoidal components, with amplitude $b_j$ and frequency $\omega_j$. The phases of the sinusoidal components are encapsulated by $t_j^*$, which is the time in months corresponding to the first maximum in the cosine function following the zero-time reference for the $j^{th}$ harmonic component, referred to below as the phase-time. The zero-time reference was chosen as the first December in the time series at each station. This aids





the pictorial representation of the phase angles of the harmonic components, with the months of the phase-time corresponding to the hours of a clock face, with December at 12 O'clock (see Figure 6 in Section 4.1). Guided by the identification of the annual cycle and its second and third harmonics in the FFT analysis discussed above, three sinusoidal terms were employed ($J$=3), with periods of 12, 6 and 4 months corresponding to frequencies $\omega$ of 1/12, 1/6 and 1/4 cycles per month, respectively.

An example of the Fourier decomposition of one of the aerosol time series - in this case from Tinga Tingana - is shown in Figure 5. The sinusoidal components are shown in the lower panel, with their respective phase-times marked by arrows. Their sum (together with the offset and linear term further discussed below) results in the fit depicted by the orange line, representing the optimal fit to the data (shown here by the climatology in black) derived using a least-squares optimization provided within the **R** language. Clearly, the fitting using the three component sinusoid is capable of representing the dual peaks in aerosol optical depth observed at Tinga Tingana and in other in arid zone stations, the peaks occurring in the spring and summer. Although the three-term sinusoidal fit has insufficient high-frequency content to represent the narrow peaks observed at this site, the higher frequency components required are not found to be continent-wide, and the use of this three-component harmonic analysis is sufficient for classification of the various aerosol regimes as further discussed below.

A list of the fitted parameters for all sites is given in Table 2. As well as the parameters defined in Equation 2, the tables lists the square of Pearson's correlation coefficient ($R^2$). The linear trend parameter $a_1$ and its significance is further discussed below.

# 4   Results

## 4.1   Amplitude and phase of the aerosol optical depth of harmonic components

A geographic depiction of the fitted amplitudes and phases at all sites is shown in Figure 6. This clearly shows the increase in amplitude of the periodic components from south to north, and the general coherence of their phase relationship across the continent. However, at Darwin the phase-time of the annual cycle is noticeably earlier (10.3 months) than at all other tropical sites (11.0 months), suggesting an earlier onset of dry season at Darwin than at other sites. Comparison of the fitted functions with those at other tropical stations (Broome, Lake Argyle and Jabiru) indeed shows slightly earlier onset of the rise toward the October peak at Darwin than at the other tropical stations. In addition, the other stations show a broader peak declining later than at Darwin. This earlier decline from the October peak contributes to earlier phase-time at Darwin than at the other sites. Both effects can be understood as the high population density around Darwin relative to the other stations, leading to many fires early in the dry season. By contrast, the burning in the more remote regions is more driven by natural ignition, and hence peaks later when the fuel is drier. The reduced influence of human activity - with fewer late season fires being lit closer to Darwin - supports the persistence of the smoke aerosol later in the dry season at the more remote stations.

A regional phase change is also evident in south-eastern Australia, in the west-east transect Adelaide, Mildura, Wagga Wagga and Canberra. The corresponding phase-times are 10.4, 11.0, 0.1 and 0.6 months respectively, showing a trend toward a later spring-summer aerosol peak in moving from west to east. This is caused by the increasing strength of the summer component relative to the spring component in the compound peak, a consequence of the increasingly forested and (hence) bushfire-prone



character of the more easterly sites. This is particularly true of Wagga Wagga and Canberra, where major incursions of smoke aerosol are found in the austral summer months. In contrast, the arid zone aerosol has a greater magnitude during the austral spring.

Aerosol sources contributing to the noted rise during spring are heterogeneous, with suggested contributions from fine aeolian dust (Mitchell et al., 2010), a combination of biomass burning smoke, fine dust, and maritime aerosol from long-range
transport (Radhi et al., 2010), and exogenous biomass burning aerosol originating in southern Africa or even South America (Rosen et al., 2000).

Both Geraldton ($t_{12}^*$=11.9 months) and Learmonth (11.7 months) on the Western Australian coast show an aerosol peak near the end of the calendar year, and are thus distinct from both the tropical stations further north, and the temperate stations further south. These stations lie in the significant but poorly-studied north-west pathway for dust transport out of the central arid zone
(Karlson et al., 2014), and their aerosol dynamics warrants closer study.

## 4.2 Comparison of harmonic amplitudes of Ångström exponent and aerosol optical depth

The fitting procedure described above was applied to the time series of both aerosol optical depth and Ångström exponent. For the Ångström exponent, this leads to an analogous set of fitted parameters to those shown in Table 2, although for brevity, these are not shown. However, once the optimised fit for each station is obtained, its peak-to-peak amplitude can be determined
for both the aerosol optical depth and Ångström exponent. A scatterplot of the amplitudes found for the Ångström exponent against that for the aerosol optical depth is shown in Figure 7, for all stations where the data record is 5 years or more.

Some separation of different aerosol regimes is possible from Figure 7. In particular, the tropical stations are easily identified by their large aerosol optical depth amplitude ($A_\tau$ >0.2) and clustering around Ångström exponent amplitude ($A_{AE} \sim 1.0$). A large number of stations lie within $0.03 < A_\tau <0.1$ but with a large range of $A_{AE}$ between 0.4 and 0.9. This rectangular
region was divided into 'Temperate' and 'Arid', with the suggested division at $A_{AE} \sim$0.6. Perhaps surprisingly given its coastal location, this classification places Adelaide in the Arid regime, possibly a result of the transport of dust over Adelaide during the summer months. By contrast, the more easterly stations at Wagga Wagga and Canberra have lower $A_{AE}$, as does Kalgoorlie, in temperate inland Western Australia.

The three stations Tennant Creek, Learmonth and Rockhampton fall outside the suggested classification rectangles. Tennant
Creek is clearly a tropical station with somewhat reduced $A_\tau$, consistent with its location between Darwin and Alice Springs. Learmonth combines the $A_{AE}$ of a tropical station with the $A_\tau$ of an Arid zone station, while Rockhampton shows a puzzlingly small $A_\tau$ given its location in coastal tropical Queensland. The aerosol regimes at Learmonth and Rockhampton warrant further study.

## 4.3 The 6 and 4 month components of the aerosol optical depth

For the tropical stations, the seasonal aerosol cycle is largely controlled by the 12-month component, with the second and third harmonics of lesser importance. However, for the arid zone and temperate stations, the higher frequency components are crucial in defining the dual-peaked nature of the compound spring-summer aerosol maximum, particularly the balance between





the spring and summer components. For example, the seasonal cycle of aerosol at the arid zone station Tinga Tingana shown in Figure 5 has dual peaks in September and January, which can be understood by the interaction of the 6 and 4 month cycles with the annual cycle.

This is further explored in Figure 8, which plots the amplitude ratio of the 4-month to 12-month component ($A_{4:12}$), against that of the 6-month to 12-month component ($A_{6:12}$). This plot exhibits an reversed 'L'-shaped distribution, with a horizontal arm at $A_{4:12} \sim 0.2$, and vertical arm at $A_{6:12} \sim 0.4$. The horizontal arm contains all the tropical stations excepting Lake Argyle, confirming the weakness of the 4-month component at these stations. At the far left of this arm lie the two west coastal stations of Geraldton and Learmonth, suggesting that their seasonal cycle is largely determined by the 12-month component, with weak contributions at both 6 and 4 months.

The vertical arm contains a mixture of arid and temperate stations, increasing toward Wagga Wagga and Canberra, the only stations where $A_{4:12}$ exceeds $A_{6:12}$. This is a consequence of the large summer smoke aerosol signal associated with these stations, as already discussed. The 'outlier' of Mildura arises because of the relatively small amplitude of the annual component, rather than excessively high amplitudes at 6 and 4 months.

## 4.4 The balance between episodic and periodic aerosol

The residual or difference between the optimised model fit to the observed time series (equation 2) and the data itself is considered to be the 'episodic' component of the aerosol signal, in the sense that it is determined by unpredictable episodes such as fires or dust storms which depart from the 'periodic' component that govern the parameters of the fitted model. Figure 9 shows the standard deviation of the residual (expressed as a fraction of the peak-to-peak amplitude of its periodic variation) against the peak-to-peak amplitude of the periodic function fitted to the aerosol optical depth at 500 nm. The standard deviation was chosen as it includes extreme values, unlike other measures of variation such as the inter-quartile range.

Figure 9 allows separation of (a) the four tropical stations (b) a cluster of 5 stations with episodic variation at $\sim 20\%$, and (c) an extended series of stations with episodic variation between 24% up to 50%. The stations with episodic variation above 30% include the smoke-affected Wagga Wagga (46%) and Canberra (49%), and the dust-dominated Tinga Tingana (36%). Geraldton (39%) and Adelaide (32%) are also in this group, although the aerosol sources responsible are less clear. This result suggests that proximity to forested areas prone to irregular bushfires (unlike the savanna burning in the tropical north) engenders the greatest episodic variation found across the sites studied.

## 4.5 Trends in aerosol optical depth

Trends in aerosol optical depth are difficult to discern because of the inherent stochastic variability imposed on the periodic signal. Hence long time series are necessary to discern a small signal amidst much noise. For the present data set, we limit the trend analysis to those time series longer than 10 years. This avoids the pitfall that can appear when a multi-year (but sub-decadal) regional change causes a change in aerosol loading. Such a change caused an increase in dust aerosol over the Lake Eyre Basin during the Millennium Drought (2003–2010), (Mitchell et al., 2010).





The model used in the present study does return a linear trend, as the coefficient $a_1$ listed in Table 2, together with the significance of the trend, expressed as the probability that the trend arose from a random distribution. It is standard practice to declare the trend significant if the probability of it arising randomly is 5% or less. In the present case, three of the stations fulfill this criteria, but we relax it slightly to 6% to enable inclusion of a fourth station. The four stations satisfying both duration of record and significance criteria are listed in Table 3.

The table shows that the trends are negative but small. For Lake Argyle and Broome in Australia's north west (Kimberley) region, the trend expressed as a change in aerosol optical depth at 500 nm is -0.002 per year, with a standard uncertainty of ±0.0008. Assuming the uncertainty is normally distributed, there is a 67% probability that the trend lies between -0.0028 and -0.0012 per year. For Rockhampton and Alice Springs, the best estimate of the trend is -0.0009 per year, with a corresponding range of -0.0012 to -0.0005. In summary, it can be stated that there is evidence for a small decrease in aerosol loading at these

four sites, with a stronger (but still small) negative trend over the Kimberley stations.

    Further context for these trends emerges from a consideration of the projected decrease in global aerosol following from scenarios under which future anthropogenic aerosol (AA) emissions decline. Rotstayn et al. (2013) modelled the impact of one such projected decline, following 'representative concentration pathway' 4.5 (RCP4.5, Taylor et al., 2012). Rotstayn et al. (2013) found the expected 'unmasking' of greenhouse warming accompanying reduced aerosol emissions, with a less-expected

increase in global rainfall caused by changed atmospheric dynamics. RCP4.5 is based on highly spatially inhomogeneous reduction in AA, with reductions over the centres of industrial activity in the northern hemisphere reflected as a large reduction in mid-visible aerosol optical depth over the northern hemisphere of 0.2 per century over the period 2006–2100. The projected decrease over the southern hemisphere is generally smaller, with a decline of <0.05 per century over Australia. Hence, the observed trends of between 0.001–0.002 per year (0.1–0.2 per century) are larger than the modelled decrease (RCP4.5) by

a factor between 2 and 4. This suggests that, over Australia, changes in natural aerosol are likely to dominate over inter-hemispheric transport of AA, hence adding weight to the climatological aerosol characterisation presented here.

## 4.6    Aerosol climatologies

Aerosol climatologies are presented in tabular and graphical formats. For each of the 22 stations included in this study, statistical summaries of aerosol optical depth at 500 nm and the Ångström exponent were prepared in tabular form, based on all individual

determinations of the measurands. This allowed generation of quartiles and medians in addition to means. The resulting tables list the monthly mean, lower quartile, median, and upper quartile for both aerosol optical depth and Ångström exponent, for all 22 stations listed in order of decreasing latitude, and shown in Table A1.

    For sites where the data record is over 5 years in length, the monthly climatology of aerosol optical depth at 500 nm is displayed in Figure 10. The time axis runs from July to June (rather than January to December) so as not to bisect the aerosol

peak, which occurs in the austral spring-summer period (September to February). The grouping of stations into regions was based on the classification suggested by Figure 7, with the three sites lying outside the rectangles labelled 'Unclassified', and Adelaide included with the 'Temperate' sites, both in view of its transitional characteristics, and to reduce clutter in the 'Arid Zone' panel. The scale length corresponding to unit optical depth is the same in all four panels to facilitate intercomparison.



### 4.7 Correlation among aerosol time series

The groupings adopted in Figure 10 are used to examine correlation of the station time series within groups. The resulting values of Pearson's correlation coefficient between each station pair in a group are shown in Tables 4–7. Unsurprisingly, given the tight clustering found in much of the foregoing analysis, the tropical group shows high correlation between stations, with even the distant pair Broome–Jabiru yielding $R$=0.8 (Table 4). Although the correlation between arid zone station pairs is weaker across the board, the pair Alice Springs–Birdsville showed the highest coefficient in the present study, at $R$=0.92 (Table 5). The large episodic aerosol component in the bushfire-prone part of the Temperate group reduces expectations of high correlation coefficients, so it is surprising to note the coefficient of $R$=0.795 between Wagga Wagga and Canberra (Table 6), suggesting regional-scale linkage via correlated smoke episodes. The results seen in Table 7 for the 'Unclassified' stations are yet more surprising, particularly the coefficient of $R$=0.73 between the pair Learmonth–Rockhampton, located on opposite sides of the continent at a distance of ∼3730 km. The implication is that the mechanisms responsible for aerosol generation and transport have much in common at both regional and trans-continental scales.

A further mechanism underlying this correlation is the inter-continental transport of aerosol plumes from southern Africa and possibly South America considered by Rosen et al. (2000). Although they measured the plumes from only one site, such material is likely to be well-dispersed following long-range transport, and was shown to contribute ∼0.02 to the mid-visible aerosol optical depth over Mildura. This is significant given the aerosol 'offset' of 0.05–0.06 typical of the arid zone stations (Table 2), so the widespread presence of this aerosol source could well boost the correlation among widely separated stations. Further analysis of the prevalence and uniformity of these layers is required.

## 5 Summary and conclusions

Analysis of the aerosol records at 22 stations across the Australian continent has been carried out with dual aims in mind: (a) generation of benchmark aerosol climatologies across the continent, and (b) classification of Australian continental aerosol types based on their measured time series. The latter goal was achieved by Fourier analysis which suggested that most of the seasonal variation could be explained by just three periodic terms, corresponding to the annual cycle and its second and third harmonics. Fitted coefficients arising from a model constructed on this basis support the following principal conclusions.

1. The proposed model explains 50% or more of the variance at 17 of the 22 sites (77%), and hence provides a basis on which to infer aerosol classification and to examine the time series for possible trends.

2. Analysis of the fitted parameters suggests that Australian aerosol can be classified into three groups: Tropical, Arid and Temperate, although there is overlap between the latter two groups on some measures.

3. The tropical class is distinguished by a dominant annual mode, resulting from savanna burning in Australia's tropical zone. Consideration of the phase of this annual mode allows the separation of different burning regimes, as seen in the earlier onset of burning near Darwin relative to the more remote tropical stations.




4. Arid zone aerosol is seasonally bimodal, with dual peaks in the austral spring and summer (typically September - January). This is evident in the increase in amplitude of the 4-monthly harmonic relative to the annual (12-month) signal.

5. Temperate stations in this study were influenced by smoke aerosol in a manner that increased with proximity to the Eucalypt forests of the Great Dividing Range. This is evinced as an increase in the episodic aerosol component, derived from the residual between the fitted (periodic) model and the measured data.

6. Significant decadal trends in aerosol optical depth at the 6% level or below were found at 4 of the 22 stations. The magnitude of the trends was small, with a decline in aerosol optical depth at 500 nm of $\sim$0.002 per year in the Kimberley region of the tropical north west and $\sim$0.001 per year at Alice Springs and Rockhampton. However, these trends are much larger than those projected to occur over Australia through future reductions in northern hemisphere anthropogenic aerosol emissions.

7. Strong correlation was found both among stations in the same group, and, remarkably, between stations separated by the full width of the continent. It follows that, on a monthly timescale, aerosol sources over Australia are driven by mechanisms that do not vary greatly either on regional or continental scales. There is evidence that inter-continental transport of biomass burning aerosol may play a significant role in this.

Despite the strong coherence in Australian aerosol just noted, we caution that this applies to monthly-mean data. The extent to which such coherence applies on shorter timescales is unclear, although its persistence among tropical stations was explored by Mitchell et al. (2013). Although a monthly climatology is useful for low frequency applications, including solar resource prediction and arguably climate change, it is by no means suitable for applications requiring instantaneous aerosol fields such as the atmospheric correction of remote sensing imagery from satellites. Hence future work will consider the relation between the instantaneous and monthly mean aerosol, and the predictive capacity of the latter. Future work will also be directed to filling in the geographical gaps evident in the present study, particularly the lack of stations over Victoria and Tasmania.

*Acknowledgements.* The CSIRO network was established with the support of the CSIRO Earth Observation Centre, with valued contributions from Dean Graetz and Denis O'Brien. We also thank the Bureau of Meteorology observing office staff and the station managers at CSIRO sites, in particular Greg Smith at Lake Argyle, Neale McShane at Birdsville, Robert Thorn at Jabiru, and Keith Leggett and Garry Dowling at Fowlers Gap. The CSIRO component of this work is supported by the Earth Observation Informatics Future Science Platform (EOI-FSP) led by Dr Arnold Dekker.





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





**Table 1.** Details of the 22 stations considered in this study.

| Station | State | Agency | Latitude | Longitude | First | Last | Months |
|---|---|---|---|---|---|---|---|
| Darwin | NT | Bureau | -12.42 | 130.89 | Apr 1999 | Dec 2015 | 194 |
| Jabiru | NT | CSIRO | -12.66 | 132.89 | Jun 2000 | Jun 2016 | 151 |
| Lake Argyle | WA | CSIRO | -16.11 | 128.75 | May 1999 | May 2016 | 177 |
| Broome | WA | Bureau | -17.95 | 122.24 | Dec 1998 | Dec 2015 | 181 |
| Townsville | QLD | Bureau | -19.25 | 146.77 | Sep 2012 | Sep 2015 | 37 |
| Tennant Creek | NT | Bureau | -19.64 | 134.18 | Jun 1999 | Jun 2006 | 82 |
| Learmonth | WA | Bureau | -22.24 | 114.10 | Dec 1998 | Dec 2015 | 87 |
| Rockhampton | QLD | Bureau | -23.38 | 150.48 | May 2000 | Dec 2015 | 183 |
| Longreach | QLD | Bureau | -23.44 | 144.28 | Dec 2012 | Jun 2015 | 31 |
| Alice Springs | NT | Bureau | -23.80 | 133.89 | Jan 2001 | Dec 2015 | 180 |
| Birdsville | QLD | CSIRO | -25.90 | 139.35 | Aug 2005 | Apr 2016 | 115 |
| Geraldton | WA | Bureau | -28.80 | 114.70 | Nov 1998 | Dec 2015 | 115 |
| Tinga Tingana | SA | CSIRO | -28.98 | 139.99 | Jul 1997 | Feb 2012 | 121 |
| Kalgoorlie | WA | Bureau | -30.78 | 121.45 | Jan 2003 | Dec 2015 | 79 |
| Fowlers Gap | NSW | CSIRO | -31.09 | 141.70 | Apr 2013 | Apr 2016 | 34 |
| Woomera | SA | Bureau | -31.16 | 136.81 | Jan 2012 | Jun 2015 | 42 |
| Lake Lefroy | WA | CSIRO | -31.25 | 121.70 | Jun 2012 | May 2016 | 46 |
| Cobar | NSW | Bureau | -31.54 | 145.80 | May 2012 | Jun 2015 | 38 |
| Mildura | Vic | Bureau | -34.24 | 142.09 | Nov 1998 | Jan 2015 | 102 |
| Adelaide | SA | Bureau | -34.95 | 138.52 | Mar 2003 | Dec 2015 | 154 |
| Wagga | NSW | Bureau | -35.16 | 147.46 | Jan 2000 | Dec 2015 | 190 |
| Canberra | ACT | CSIRO | -35.27 | 149.11 | Jan 2003 | Jul 2016 | 148 |





**Table 2.** Parameters of the fits to the aerosol optical depth time series derived from the 22 stations. $R^2$ is the square of Pearson's correlation coefficient of the fit between the model given in Equation 2 and the aerosol optical depth time series at 500 nm. The parameters $a_0$ and $a_1$ give the AOD offset and linear coefficient respectively, the latter in units of AOD per year. The following column tabulates the significance of the trend, expressed as the probability that the trend arises from a random distribution. Trends were considered significant for $P(a_1)$ of <6%, indicated in bold face. Remaining columns list the coefficients $b$ and phase for the three sinusoidal components with periods of 12 months, 6 months and 4 months, respectively. The phase is expressed as the time in months at which the basis cosine function reaches its first maximum. The station labelled Canberra[b] in the last row of the table demonstrates the improvement in the fit when the effects of the firestorm of January 2003 are excised. Exclusion of the months January, February, March and April 2003 leads to an increase in the value of $R^2$ from 0.181 to 0.413.

| Station | Months | $R^2$ | $a_0$ | $a_1$ | $P(a_1)$ | 12 month | | 6 month | | 4 month | |
| | | | | | | $b_{12}$ | $t_{12}$ | $b_6$ | $t_6$ | $b_4$ | $t_4$ |
|---|---|---|---|---|---|---|---|---|---|---|---|
| Darwin | 194 | 0.670 | 0.150 | 0.000 | 0.669 | 0.079 | 10.3 | 0.028 | 3.8 | 0.013 | 1.8 |
| Jabiru | 151 | 0.685 | 0.165 | -0.001 | 0.198 | 0.089 | 11.0 | 0.026 | 4.2 | 0.016 | 1.8 |
| Lake Argyle | 177 | 0.601 | 0.147 | **-0.002** | **0.056** | 0.086 | 11.0 | 0.037 | 4.0 | 0.023 | 1.5 |
| Broome | 181 | 0.527 | 0.136 | **-0.002** | **0.005** | 0.067 | 11.0 | 0.029 | 4.0 | 0.013 | 1.7 |
| Townsville | 37 | 0.604 | 0.094 | -0.006 | 0.168 | 0.031 | 11.3 | 0.005 | 3.1 | 0.011 | 1.3 |
| Tennant Creek | 82 | 0.688 | 0.090 | -0.004 | 0.038 | 0.054 | 10.9 | 0.022 | 3.9 | 0.011 | 1.7 |
| Learmonth | 87 | 0.736 | 0.055 | 0.001 | 0.000 | 0.033 | 11.7 | 0.009 | 3.4 | 0.006 | 1.6 |
| Rockhampton | 183 | 0.692 | 0.087 | **-0.001** | **0.015** | 0.043 | 11.4 | 0.012 | 3.8 | 0.005 | 1.4 |
| Longreach | 31 | 0.697 | 0.072 | -0.011 | 0.046 | 0.036 | 11.4 | 0.009 | 4.6 | 0.003 | 1.6 |
| Alice Springs | 180 | 0.586 | 0.064 | **-0.001** | **0.021** | 0.033 | 11.2 | 0.011 | 3.8 | 0.006 | 1.7 |
| Birdsville | 115 | 0.688 | 0.056 | -0.001 | 0.100 | 0.024 | 10.9 | 0.010 | 3.6 | 0.007 | 1.6 |
| Geraldton | 115 | 0.397 | 0.064 | -0.000 | 0.365 | 0.031 | 11.9 | 0.005 | 2.1 | 0.006 | 1.3 |
| Tinga Tingana | 121 | 0.382 | 0.048 | 0.001 | 0.157 | 0.023 | 11.4 | 0.011 | 2.9 | 0.011 | 1.1 |
| Kalgoorlie | 79 | 0.737 | 0.046 | 0.000 | 0.686 | 0.016 | 11.0 | 0.007 | 3.1 | 0.006 | 1.3 |
| Fowlers Gap | 34 | 0.731 | 0.042 | 0.004 | 0.010 | 0.011 | 10.9 | 0.006 | 3.0 | 0.006 | 1.6 |
| Woomera | 42 | 0.752 | 0.040 | 0.002 | 0.044 | 0.014 | 10.7 | 0.007 | 3.6 | 0.006 | 1.6 |
| Lake Lefroy | 46 | 0.756 | 0.049 | 0.002 | 0.026 | 0.016 | 11.1 | 0.005 | 2.8 | 0.007 | 1.3 |
| Cobar | 38 | 0.750 | 0.045 | -0.001 | 0.315 | 0.013 | 11.4 | 0.005 | 2.9 | 0.009 | 1.5 |
| Mildura | 102 | 0.476 | 0.056 | -0.000 | 0.053 | 0.011 | 11.0 | 0.009 | 2.9 | 0.008 | 1.3 |
| Adelaide | 154 | 0.510 | 0.058 | 0.000 | 0.880 | 0.014 | 10.4 | 0.006 | 3.3 | 0.006 | 1.2 |
| Wagga Wagga | 190 | 0.310 | 0.061 | -0.000 | 0.944 | 0.017 | 0.1 | 0.007 | 2.9 | 0.008 | 1.2 |
| Canberra | 148 | 0.181 | 0.077 | -0.002 | 0.131 | 0.025 | 0.6 | 0.009 | 2.3 | 0.012 | 1.1 |
| Canberra[b] | 144 | 0.413 | 0.061 | 0.000 | 0.308 | 0.018 | 0.4 | 0.009 | 3.1 | 0.005 | 1.2 |





**Table 3.** Analysis of linear trends $a_1$ expressed as the change in aerosol optical depth per year. Only stations with at least a 10-year data record and trend significance of P($a_1$)≤6% are listed. $\sigma(a_1)$ is the standard uncertainty in $a_1$.

| Station | Months | P($a_1$) | $a_1$ (yr$^{-1}$) | $\sigma(a_1)$ |
|---|---|---|---|---|
| Lake Argyle | 177 | 0.056 | -0.0017 | 0.00086 |
| Broome | 181 | 0.005 | -0.0022 | 0.00077 |
| Rockhampton | 183 | 0.015 | -0.0009 | 0.00035 |
| Alice Springs | 180 | 0.021 | -0.0009 | 0.00037 |

**Table 4.** Pearson's correlation coefficient between all Tropical station pairs, for stations with time series of 60 months or greater.

| Tropical Station | Darwin | Jabiru | L. Argyle | Broome |
|---|---|---|---|---|
| Darwin | 1.000 | 0.888 | 0.827 | 0.750 |
| Jabiru | 0.888 | 1.000 | 0.869 | 0.804 |
| L. Argyle | 0.827 | 0.869 | 1.000 | 0.883 |
| Broome | 0.750 | 0.804 | 0.883 | 1.000 |

**Table 5.** Pearson's correlation coefficient between all Arid Zone station pairs, for stations with time series of 60 months or greater.

| Arid Zone Station | Alice | Birdsville | Geraldton | T. Tingana | Mildura |
|---|---|---|---|---|---|
| Alice | 1.000 | 0.920 | 0.569 | 0.732 | 0.577 |
| Birdsville | 0.920 | 1.000 | 0.550 | 0.748 | 0.574 |
| Geraldton | 0.569 | 0.550 | 1.000 | 0.361 | 0.469 |
| T. Tingana | 0.732 | 0.748 | 0.361 | 1.000 | 0.757 |
| Mildura | 0.577 | 0.574 | 0.469 | 0.757 | 1.000 |

**Table 6.** Pearson's correlation coefficient between all Temperate station pairs, for stations with time series of 60 months or greater.

| Temperate Station | Kalg. | Adel. | Wagga | Canberra |
|---|---|---|---|---|
| Kalg. | 1.000 | 0.680 | 0.492 | 0.263 |
| Adel. | 0.680 | 1.000 | 0.575 | 0.375 |
| Wagga | 0.492 | 0.575 | 1.000 | 0.795 |
| Canberra | 0.263 | 0.375 | 0.795 | 1.000 |





**Table 7.** Pearson's correlation coefficient between all unclassified station pairs, for stations with time series of 60 months or greater.

| Unclassified Station | Tennant Ck | Learmonth | Rockhampton |
|---|---|---|---|
| Tennant Ck | 1.000 | 0.761 | 0.829 |
| Learmonth | 0.761 | 1.000 | 0.726 |
| Rockhampton | 0.829 | 0.726 | 1.000 |





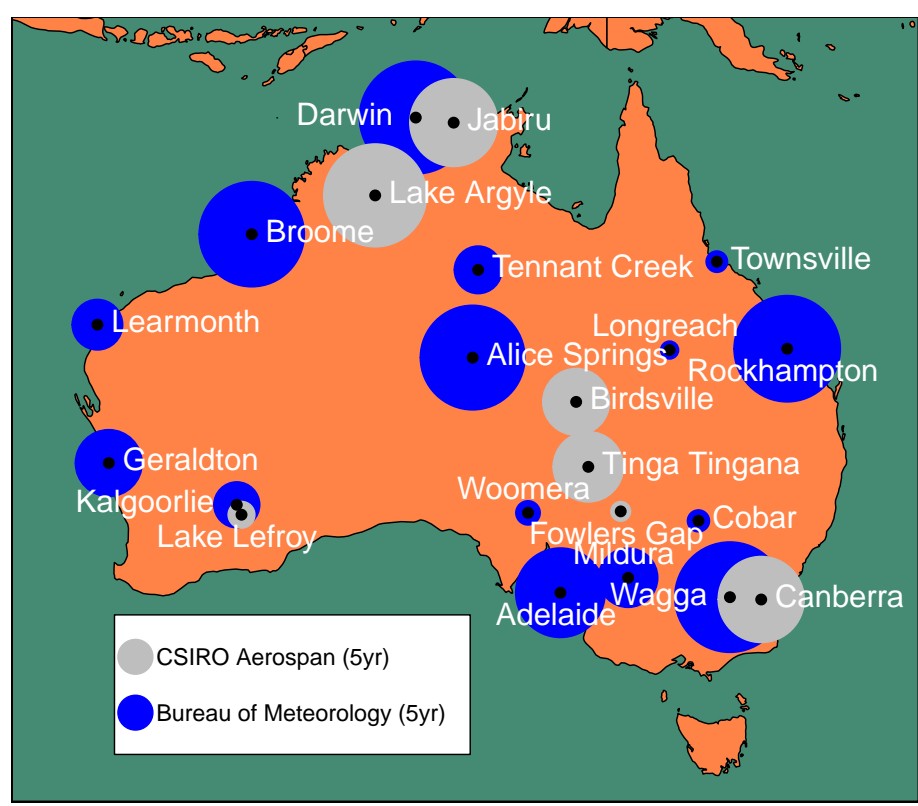

**Figure 1.** Location of sun photometer stations in Australia. Bureau of Meteorology stations are shown in blue, and CSIRO/AeroSpan stations in grey. The radius of the filled circles scales as the duration of the data record at a given station. The filled circles shown in the legend are for a duration of 5 yr.





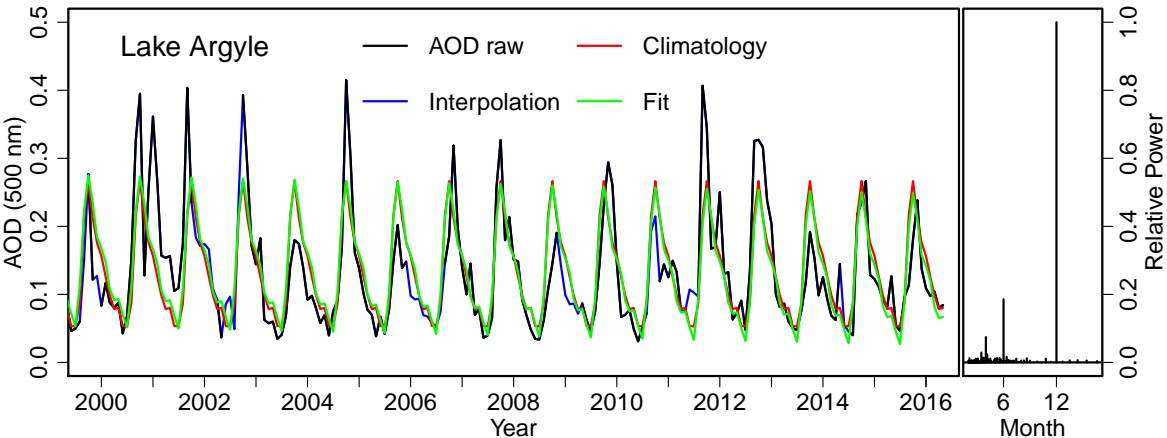

**Figure 2.** Left panel: Time series of monthly mean aerosol optical depth at 500 nm at Lake Argyle. The raw data is shown in black, the (repeated) climatology in red, the interpolated data in blue (see text), and the model fit is shown in green. Right panel: The power spectrum of the interpolated time series computed using a fast Fourier transform (FFT).

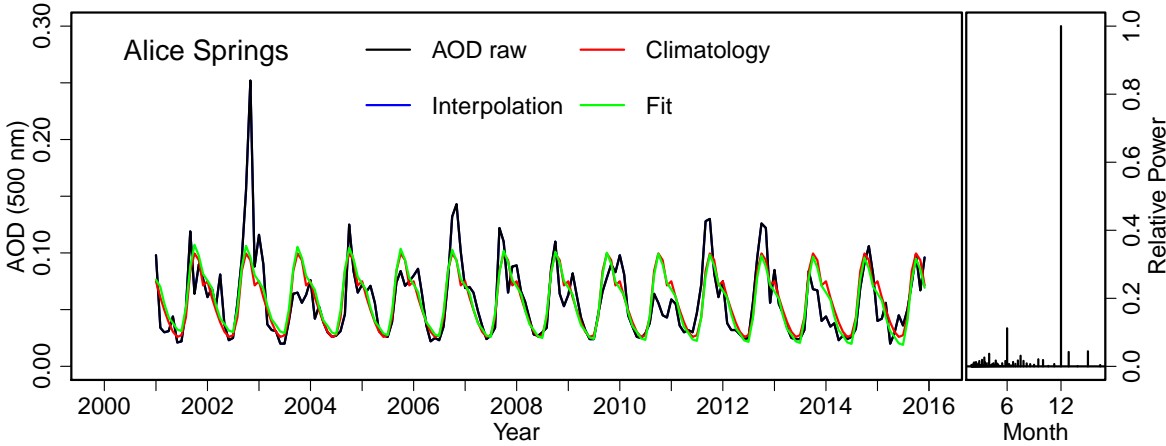

**Figure 3.** Left panel: Time series of monthly mean aerosol optical depth at 500 nm at Alice Springs. The raw data is shown in black, the (repeated) climatology in red, the interpolated data in blue (see text), and the model fit is shown in green. Right panel: The power spectrum of the interpolated time series computed using a fast Fourier transform (FFT).

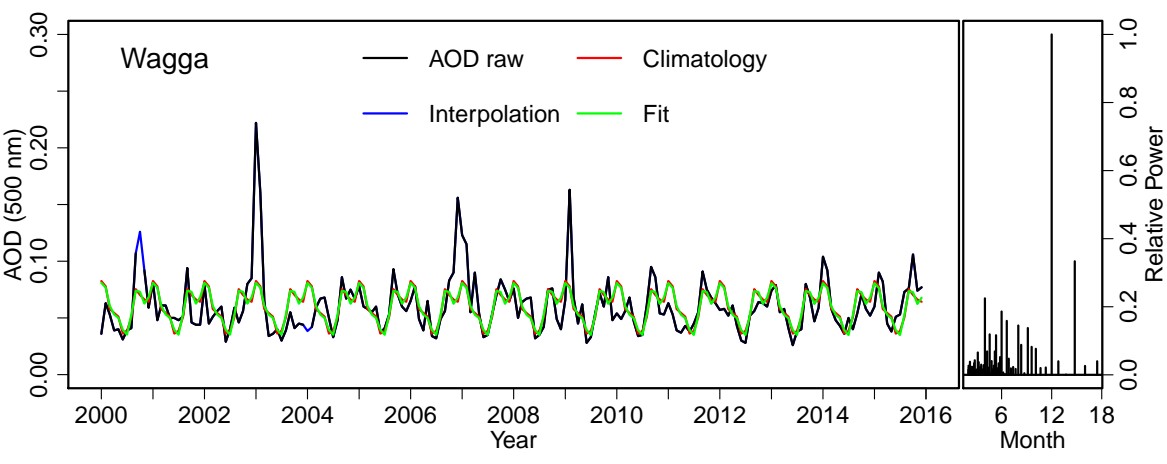

**Figure 4.** Left panel: Time series of monthly mean aerosol optical depth at 500 nm at Wagga Wagga. The raw data is shown in black, the (repeated) climatology in red, the interpolated data in blue (see text), and the model fit is shown in green. Right panel: The power spectrum of the interpolated time series computed using a fast Fourier transform (FFT).





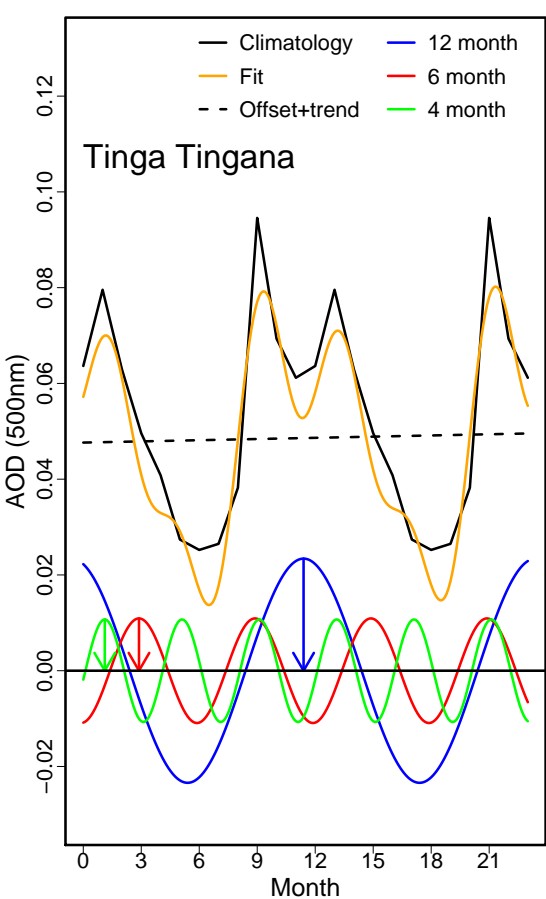

**Figure 5.** Illustration of the Fourier decomposition of the periodic component of the aerosol optical depth at Tinga Tingana. The fitted curve (orange) results from the addition of the offset plus linear trend (dashed line) to the three sinusoids corresponding to periods of 12 months (blue), 6 months (red) and 4 months (green). The arrowed phase-times of the three sinusoids were 11.4, 2.9 and 1.1 months respectively. The phase-times are referenced to the first December in each time series, denoted as month zero in this plot. The observed climatology is shown as the black solid line. The linear trend apparent in the dashed line is not significant at the 5% level (see Table 2).




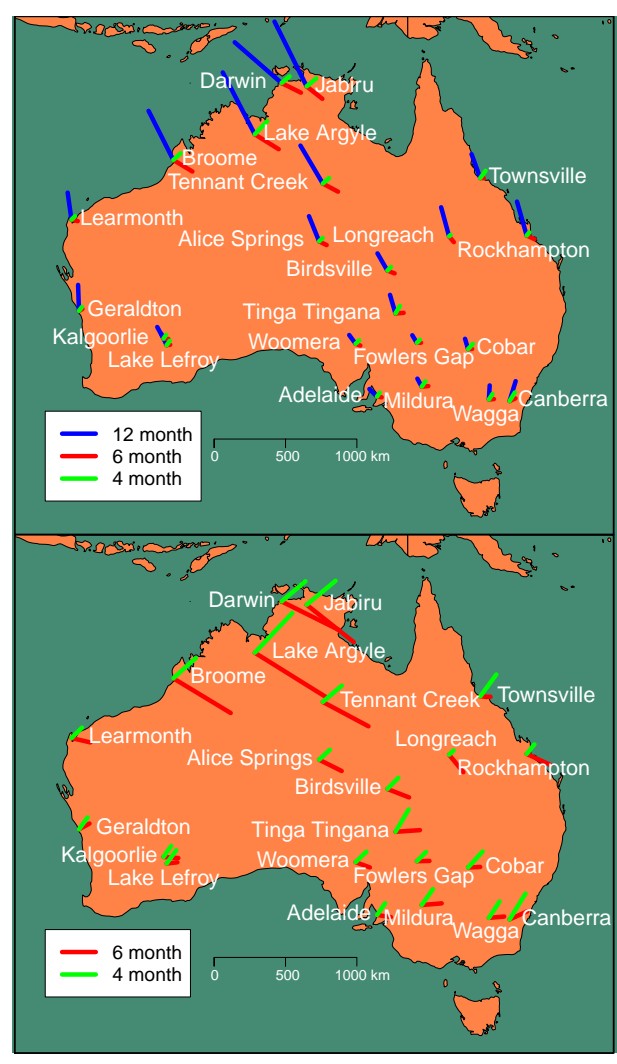

**Figure 6.** Amplitude and phase of the 12-month (blue), 6-month (red) and 4-month (green) components of the sinusoidal functions fitted to the time series of aerosol optical depth at 500 nm. The length of the bars is proportional to their amplitude, while the angles measured clockwise from due north indicate their phase. In the upper panel, all three components are shown. The lower panel presents an expanded version of the 6-month (red) and 4-month (green) components, where the lengths of these two components are scaled up by a factor of 3 relative to the upper panel.





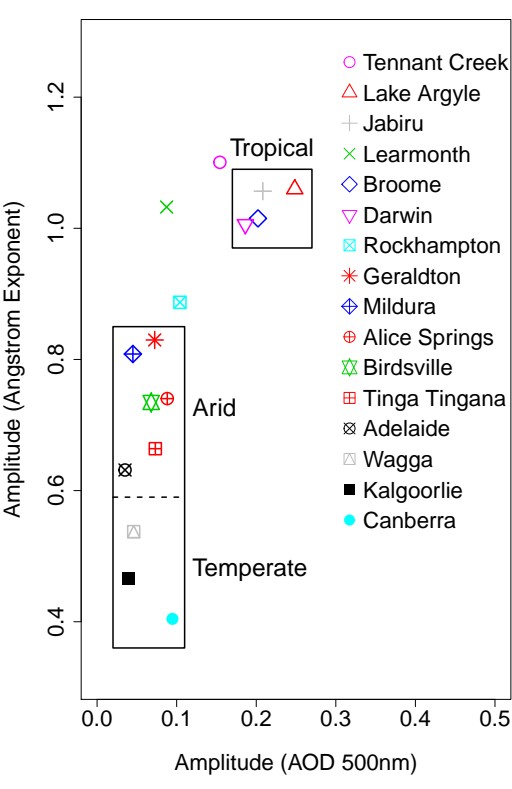

**Figure 7.** Plot of the periodic amplitudes of the Ångström exponent against aerosol optical depth at 500 nm. The amplitudes were obtained from the peak-to-peak excursion of the fitted functions representing the periodic component of the respective aerosol quantities.





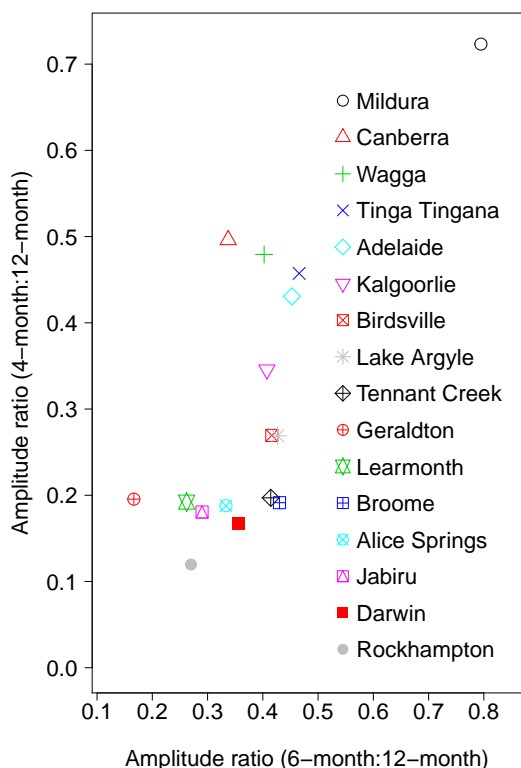

**Figure 8.** Plot of the amplitudes of the sinusoidal components with a period of 4 months (vertical axis) against the amplitude of the 6 monthly component (horizontal axis), both relative to the 12-month amplitude. The relative 4-monthly amplitude controls the extent of double-peaking in the spring-summer period.





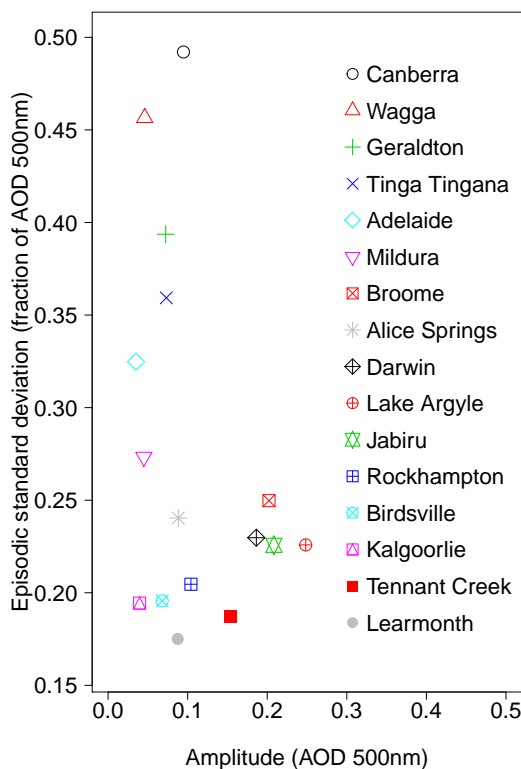

**Figure 9.** The standard deviation of the episodic component of the aerosol optical depth expressed as a fraction of the amplitude of the periodic aerosol optical depth at 500 nm, plotted against the amplitude of the periodic aerosol optical depth at 500 nm. This allows clear separation between (a) the tropical stations, and (b) stations effected by increasingly large episodic activity. Stations where the episodic standard deviation is more than 30% of the periodic component include the smoke-affected temperate stations of Wagga Wagga and Canberra, and the dust-dominated station at Tinga Tingana. The high episodic aerosol at Wagga Wagga and Canberra is consistent with the increasing incidence of smoke aerosol from forested areas from west to east.





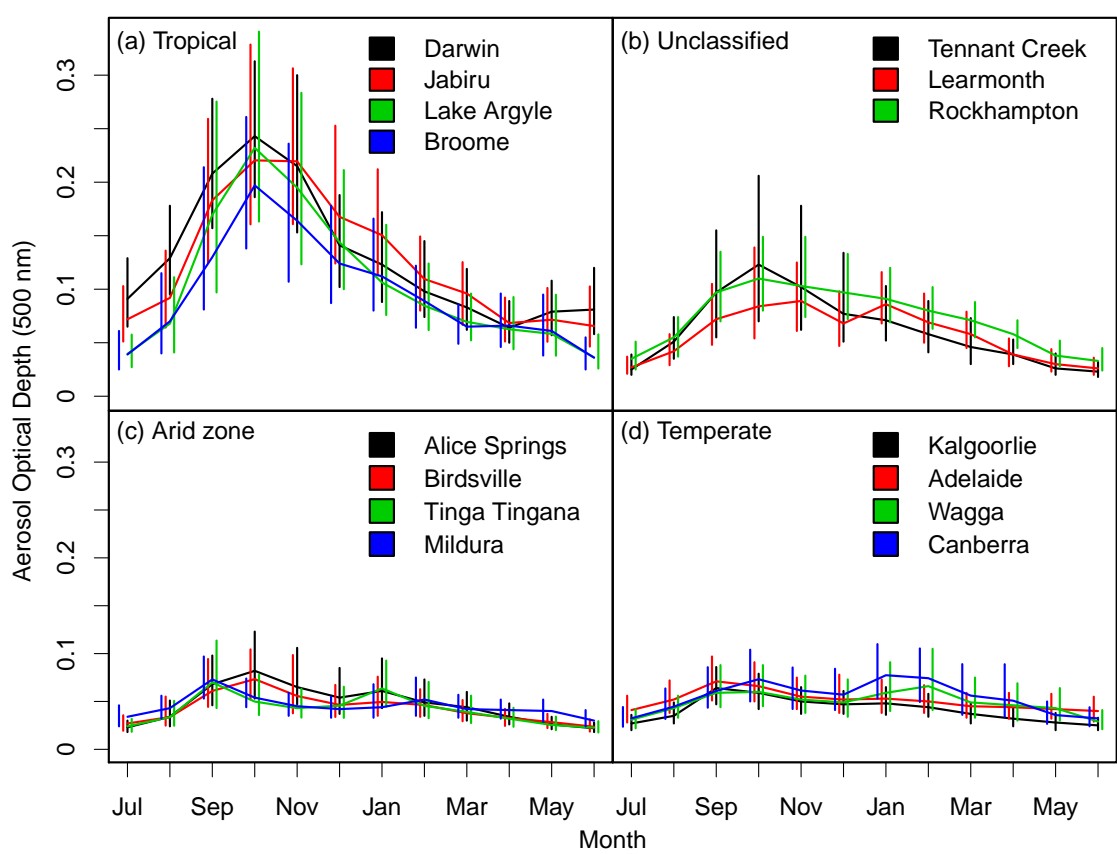

**Figure 10.** The monthly climatology of aerosol optical depth at 500 nm, for (a) Tropical, (b) Unclassified, (c) Arid zone, and (d) Temperate regions. The plots depict the monthly median, with the vertical bars showing the inter-quartile range, i.e., the range in aerosol optical depth between the 25th and 75th percentiles. The vertical bars are offset between sites for clarity.





# Appendix A: Climatological tables

**Table A1.** Monthly climatology of 500 nm aerosol optical depth and Ångström exponent at all 22 stations, listed in order of decreasing latitude. The columns labelled 25Q and 75Q list the break points at the first quartile (25%) and third quartile (75%) respectively. For the Bureau stations (denoted 'BoM' in the header), the Ångström exponent before 2009 was calculated using the wavelength pair (500, 778 nm). For 2009 and beyond, the wavelength pair (500, 868 nm) was used (see text). For the CSIRO AeroSpan stations, denoted 'CSIRO', the wavelength pair (440, 870 nm) was used throughout.

| Darwin (12.42 S, BoM) | | | Aerosol Optical Depth, 500 nm | | | | | Ångström exponent | | | |
|---|---|---|---|---|---|---|---|---|---|---|---|
| Mon | Yrs | Nobs | Mean | 25Q | Median | 75Q | Nobs | Mean | 25Q | Median | 75Q |
| 1 | 13 | 17430 | 0.140 | 0.088 | 0.123 | 0.172 | 17430 | 0.57 | 0.27 | 0.54 | 0.84 |
| 2 | 12 | 21399 | 0.120 | 0.074 | 0.098 | 0.145 | 21399 | 0.58 | 0.26 | 0.68 | 1.05 |
| 3 | 13 | 31050 | 0.098 | 0.062 | 0.083 | 0.119 | 31050 | 0.86 | 0.53 | 0.89 | 1.16 |
| 4 | 12 | 73545 | 0.078 | 0.050 | 0.064 | 0.089 | 73545 | 0.93 | 0.58 | 0.92 | 1.24 |
| 5 | 13 | 105951 | 0.092 | 0.057 | 0.079 | 0.108 | 105951 | 1.13 | 0.81 | 1.15 | 1.46 |
| 6 | 14 | 134604 | 0.099 | 0.058 | 0.081 | 0.120 | 134604 | 1.22 | 0.89 | 1.22 | 1.59 |
| 7 | 14 | 139413 | 0.105 | 0.065 | 0.091 | 0.129 | 139413 | 1.18 | 0.88 | 1.20 | 1.49 |
| 8 | 15 | 126444 | 0.149 | 0.095 | 0.129 | 0.178 | 126444 | 1.29 | 1.05 | 1.30 | 1.54 |
| 9 | 15 | 108975 | 0.230 | 0.157 | 0.208 | 0.278 | 108975 | 1.34 | 1.19 | 1.36 | 1.54 |
| 10 | 15 | 86430 | 0.260 | 0.186 | 0.243 | 0.313 | 86430 | 1.26 | 1.12 | 1.30 | 1.45 |
| 11 | 15 | 54747 | 0.238 | 0.153 | 0.215 | 0.300 | 54747 | 1.22 | 1.01 | 1.29 | 1.50 |
| 12 | 15 | 36378 | 0.165 | 0.102 | 0.141 | 0.188 | 36378 | 0.90 | 0.65 | 0.96 | 1.24 |

| Jabiru (12.66 S, CSIRO) | | | Aerosol Optical Depth, 500 nm | | | | | Ångström exponent | | | |
|---|---|---|---|---|---|---|---|---|---|---|---|
| Mon | Yrs | Nobs | Mean | 25Q | Median | 75Q | Nobs | Mean | 25Q | Median | 75Q |
| 01 | 9 | 6417 | 0.187 | 0.113 | 0.150 | 0.212 | 6417 | 0.57 | 0.21 | 0.52 | 0.85 |
| 02 | 8 | 8371 | 0.128 | 0.080 | 0.110 | 0.149 | 8371 | 0.65 | 0.29 | 0.68 | 1.06 |
| 03 | 10 | 12124 | 0.114 | 0.067 | 0.096 | 0.125 | 12124 | 0.78 | 0.42 | 0.80 | 1.11 |
| 04 | 11 | 25762 | 0.084 | 0.051 | 0.068 | 0.092 | 25762 | 0.85 | 0.49 | 0.86 | 1.23 |
| 05 | 13 | 40694 | 0.088 | 0.051 | 0.071 | 0.101 | 40694 | 0.97 | 0.60 | 0.99 | 1.35 |
| 06 | 15 | 53134 | 0.088 | 0.046 | 0.066 | 0.103 | 53134 | 1.03 | 0.64 | 1.03 | 1.45 |
| 07 | 15 | 64731 | 0.085 | 0.051 | 0.072 | 0.103 | 64731 | 1.07 | 0.77 | 1.11 | 1.42 |
| 08 | 15 | 65423 | 0.117 | 0.064 | 0.092 | 0.136 | 65423 | 1.27 | 1.02 | 1.32 | 1.58 |
| 09 | 15 | 49354 | 0.212 | 0.125 | 0.183 | 0.259 | 49354 | 1.35 | 1.17 | 1.40 | 1.59 |
| 10 | 15 | 35388 | 0.259 | 0.161 | 0.220 | 0.329 | 35388 | 1.33 | 1.16 | 1.36 | 1.54 |
| 11 | 12 | 21807 | 0.250 | 0.161 | 0.220 | 0.306 | 21807 | 1.19 | 0.99 | 1.22 | 1.45 |
| 12 | 12 | 14469 | 0.214 | 0.124 | 0.168 | 0.253 | 14469 | 0.78 | 0.40 | 0.83 | 1.18 |





| Lake Argyle (16.11 S, CSIRO) | | Aerosol Optical Depth, 500 nm | | | | | Ångström exponent | | | |
|---|---|---|---|---|---|---|---|---|---|---|
| Mon | Yrs | Nobs | Mean | 25Q | Median | 75Q | Nobs | Mean | 25Q | Median | 75Q |
| 01 | 14 | 25361 | 0.146 | 0.076 | 0.106 | 0.160 | 25361 | 0.48 | 0.16 | 0.37 | 0.73 |
| 02 | 14 | 24189 | 0.114 | 0.062 | 0.085 | 0.124 | 24189 | 0.63 | 0.22 | 0.60 | 0.99 |
| 03 | 14 | 30789 | 0.090 | 0.052 | 0.070 | 0.096 | 30789 | 0.71 | 0.38 | 0.72 | 1.02 |
| 04 | 15 | 45877 | 0.077 | 0.044 | 0.063 | 0.093 | 45877 | 1.04 | 0.73 | 1.09 | 1.38 |
| 05 | 17 | 51042 | 0.078 | 0.038 | 0.058 | 0.095 | 51042 | 1.12 | 0.81 | 1.15 | 1.46 |
| 06 | 13 | 43909 | 0.050 | 0.026 | 0.036 | 0.058 | 43909 | 1.04 | 0.77 | 1.07 | 1.35 |
| 07 | 14 | 50725 | 0.051 | 0.027 | 0.040 | 0.058 | 50725 | 1.11 | 0.81 | 1.11 | 1.43 |
| 08 | 15 | 60426 | 0.088 | 0.041 | 0.068 | 0.111 | 60426 | 1.23 | 1.02 | 1.27 | 1.49 |
| 09 | 14 | 58752 | 0.212 | 0.097 | 0.170 | 0.275 | 58752 | 1.34 | 1.15 | 1.40 | 1.61 |
| 10 | 16 | 50694 | 0.266 | 0.163 | 0.232 | 0.341 | 50694 | 1.39 | 1.21 | 1.43 | 1.60 |
| 11 | 16 | 45741 | 0.219 | 0.123 | 0.195 | 0.283 | 45741 | 1.30 | 1.14 | 1.38 | 1.57 |
| 12 | 14 | 27041 | 0.175 | 0.100 | 0.144 | 0.211 | 27041 | 0.84 | 0.48 | 0.88 | 1.20 |

| Broome (17.95 S, BoM) | | Aerosol Optical Depth, 500 nm | | | | | Ångström exponent | | | |
|---|---|---|---|---|---|---|---|---|---|---|
| Mon | Yrs | Nobs | Mean | 25Q | Median | 75Q | Nobs | Mean | 25Q | Median | 75Q |
| 1 | 15 | 63441 | 0.137 | 0.080 | 0.112 | 0.166 | 63441 | 0.38 | 0.12 | 0.34 | 0.61 |
| 2 | 16 | 64785 | 0.106 | 0.064 | 0.089 | 0.122 | 64785 | 0.42 | 0.17 | 0.37 | 0.62 |
| 3 | 15 | 101973 | 0.077 | 0.049 | 0.065 | 0.086 | 101973 | 0.61 | 0.32 | 0.63 | 0.93 |
| 4 | 13 | 142407 | 0.079 | 0.046 | 0.066 | 0.096 | 142407 | 1.05 | 0.71 | 1.12 | 1.44 |
| 5 | 13 | 154413 | 0.076 | 0.038 | 0.061 | 0.095 | 154413 | 1.20 | 0.88 | 1.26 | 1.57 |
| 6 | 13 | 161550 | 0.047 | 0.025 | 0.036 | 0.055 | 161550 | 0.94 | 0.54 | 0.98 | 1.33 |
| 7 | 13 | 168627 | 0.048 | 0.025 | 0.039 | 0.061 | 168627 | 0.92 | 0.56 | 0.97 | 1.33 |
| 8 | 14 | 185094 | 0.090 | 0.040 | 0.070 | 0.115 | 185094 | 1.07 | 0.74 | 1.16 | 1.50 |
| 9 | 16 | 211056 | 0.162 | 0.081 | 0.130 | 0.214 | 211056 | 1.28 | 1.06 | 1.32 | 1.56 |
| 10 | 16 | 202263 | 0.210 | 0.138 | 0.197 | 0.261 | 202263 | 1.19 | 1.01 | 1.27 | 1.49 |
| 11 | 16 | 154101 | 0.185 | 0.107 | 0.164 | 0.236 | 154101 | 1.12 | 0.86 | 1.13 | 1.42 |
| 12 | 17 | 117633 | 0.141 | 0.087 | 0.124 | 0.178 | 117633 | 0.71 | 0.31 | 0.65 | 1.08 |





| Townsville (19.25 S, BoM) | | | Aerosol Optical Depth, 500 nm | | | | Ångström exponent | | | |
|---|---|---|---|---|---|---|---|---|---|---|---|
| Mon | Yrs | Nobs | Mean | 25Q | Median | 75Q | Nobs | Mean | 25Q | Median | 75Q |
| 1 | 3 | 11097 | 0.103 | 0.071 | 0.088 | 0.122 | 11097 | 0.63 | 0.28 | 0.62 | 0.93 |
| 2 | 3 | 11856 | 0.090 | 0.048 | 0.069 | 0.100 | 11856 | 0.71 | 0.38 | 0.74 | 0.98 |
| 3 | 3 | 16746 | 0.073 | 0.054 | 0.068 | 0.084 | 16746 | 0.38 | 0.14 | 0.38 | 0.61 |
| 4 | 3 | 18690 | 0.056 | 0.039 | 0.051 | 0.067 | 18690 | 0.65 | 0.35 | 0.58 | 0.96 |
| 5 | 3 | 21204 | 0.057 | 0.040 | 0.056 | 0.070 | 21204 | 0.50 | 0.17 | 0.44 | 0.80 |
| 6 | 3 | 18183 | 0.058 | 0.040 | 0.051 | 0.071 | 18183 | 0.72 | 0.40 | 0.72 | 1.08 |
| 7 | 3 | 25752 | 0.054 | 0.039 | 0.049 | 0.066 | 25752 | 0.59 | 0.35 | 0.62 | 0.80 |
| 8 | 3 | 33111 | 0.068 | 0.046 | 0.061 | 0.082 | 33111 | 0.84 | 0.62 | 0.86 | 1.07 |
| 9 | 4 | 33579 | 0.108 | 0.070 | 0.096 | 0.135 | 33579 | 0.91 | 0.69 | 0.94 | 1.15 |
| 10 | 3 | 29700 | 0.114 | 0.084 | 0.105 | 0.127 | 29700 | 0.86 | 0.64 | 0.87 | 1.09 |
| 11 | 3 | 21135 | 0.105 | 0.071 | 0.094 | 0.129 | 21135 | 0.76 | 0.50 | 0.73 | 1.02 |
| 12 | 3 | 22038 | 0.111 | 0.070 | 0.092 | 0.117 | 22038 | 0.74 | 0.44 | 0.69 | 1.05 |

| Tennant Creek (19.64 S, BoM) | | | Aerosol Optical Depth, 500 nm | | | | Ångström exponent | | | |
|---|---|---|---|---|---|---|---|---|---|---|---|
| Mon | Yrs | Nobs | Mean | 25Q | Median | 75Q | Nobs | Mean | 25Q | Median | 75Q |
| 1 | 7 | 53808 | 0.087 | 0.052 | 0.071 | 0.103 | 53808 | 1.02 | 0.63 | 1.04 | 1.41 |
| 2 | 7 | 46875 | 0.079 | 0.041 | 0.058 | 0.089 | 46875 | 0.79 | 0.43 | 0.73 | 1.10 |
| 3 | 7 | 62208 | 0.056 | 0.030 | 0.046 | 0.073 | 62208 | 1.22 | 0.70 | 1.10 | 1.53 |
| 4 | 7 | 74208 | 0.046 | 0.030 | 0.039 | 0.053 | 74208 | 1.28 | 0.80 | 1.12 | 1.72 |
| 5 | 7 | 78627 | 0.039 | 0.020 | 0.026 | 0.040 | 78627 | 1.17 | 0.91 | 1.19 | 1.42 |
| 6 | 7 | 73527 | 0.029 | 0.018 | 0.023 | 0.032 | 73527 | 1.00 | 0.73 | 1.02 | 1.28 |
| 7 | 7 | 98988 | 0.034 | 0.020 | 0.025 | 0.039 | 98988 | 1.16 | 0.75 | 1.09 | 1.47 |
| 8 | 7 | 104730 | 0.063 | 0.035 | 0.051 | 0.074 | 104730 | 1.30 | 1.01 | 1.30 | 1.61 |
| 9 | 7 | 80706 | 0.126 | 0.055 | 0.097 | 0.155 | 80706 | 1.24 | 1.06 | 1.29 | 1.46 |
| 10 | 7 | 81009 | 0.159 | 0.070 | 0.123 | 0.206 | 81009 | 1.44 | 1.25 | 1.45 | 1.62 |
| 11 | 7 | 59589 | 0.138 | 0.062 | 0.102 | 0.178 | 59589 | 1.22 | 0.97 | 1.30 | 1.50 |
| 12 | 7 | 54309 | 0.101 | 0.051 | 0.077 | 0.134 | 54309 | 1.22 | 0.91 | 1.23 | 1.60 |



| Learmonth (22.24 S, BoM) | | | Aerosol Optical Depth, 500 nm | | | | Ångström exponent | | | | |
|---|---|---|---|---|---|---|---|---|---|---|---|
| Mon | Yrs | Nobs | Mean | 25Q | Median | 75Q | Nobs | Mean | 25Q | Median | 75Q |
| 1 | 7 | 69039 | 0.099 | 0.068 | 0.086 | 0.116 | 69039 | 0.34 | 0.12 | 0.29 | 0.53 |
| 2 | 6 | 73647 | 0.078 | 0.050 | 0.069 | 0.096 | 73647 | 0.29 | 0.10 | 0.34 | 0.54 |
| 3 | 6 | 61080 | 0.064 | 0.045 | 0.058 | 0.079 | 61080 | 0.36 | 0.08 | 0.31 | 0.66 |
| 4 | 6 | 54321 | 0.045 | 0.028 | 0.039 | 0.054 | 54321 | 0.67 | 0.39 | 0.68 | 0.98 |
| 5 | 6 | 51885 | 0.036 | 0.023 | 0.030 | 0.044 | 51885 | 0.71 | 0.39 | 0.72 | 1.00 |
| 6 | 8 | 73611 | 0.030 | 0.020 | 0.026 | 0.036 | 73611 | 0.64 | 0.30 | 0.64 | 0.96 |
| 7 | 7 | 85224 | 0.031 | 0.021 | 0.027 | 0.037 | 85224 | 0.61 | 0.25 | 0.53 | 0.96 |
| 8 | 8 | 101604 | 0.047 | 0.029 | 0.042 | 0.058 | 101604 | 0.85 | 0.53 | 0.86 | 1.16 |
| 9 | 8 | 106053 | 0.083 | 0.048 | 0.072 | 0.105 | 106053 | 1.00 | 0.74 | 1.02 | 1.27 |
| 10 | 8 | 113100 | 0.100 | 0.054 | 0.084 | 0.139 | 113100 | 0.98 | 0.72 | 1.05 | 1.27 |
| 11 | 7 | 105447 | 0.097 | 0.061 | 0.089 | 0.125 | 105447 | 0.73 | 0.43 | 0.72 | 1.00 |
| 12 | 8 | 107043 | 0.076 | 0.047 | 0.068 | 0.098 | 107043 | 0.55 | 0.23 | 0.54 | 0.80 |

| Rockhampton (23.38 S, BoM) | | | Aerosol Optical Depth, 500 nm | | | | Ångström exponent | | | | |
|---|---|---|---|---|---|---|---|---|---|---|---|
| Mon | Yrs | Nobs | Mean | 25Q | Median | 75Q | Nobs | Mean | 25Q | Median | 75Q |
| 1 | 15 | 62601 | 0.107 | 0.069 | 0.091 | 0.120 | 60318 | 0.88 | 0.58 | 0.91 | 1.20 |
| 2 | 15 | 59529 | 0.090 | 0.063 | 0.080 | 0.102 | 57762 | 0.85 | 0.50 | 0.85 | 1.22 |
| 3 | 15 | 78348 | 0.074 | 0.055 | 0.071 | 0.088 | 77493 | 0.72 | 0.38 | 0.69 | 1.00 |
| 4 | 15 | 100890 | 0.061 | 0.045 | 0.058 | 0.071 | 99675 | 0.81 | 0.48 | 0.88 | 1.21 |
| 5 | 16 | 123183 | 0.043 | 0.028 | 0.038 | 0.052 | 115098 | 0.96 | 0.59 | 1.00 | 1.37 |
| 6 | 16 | 109758 | 0.037 | 0.024 | 0.033 | 0.045 | 94458 | 0.92 | 0.59 | 0.94 | 1.27 |
| 7 | 16 | 142692 | 0.041 | 0.025 | 0.035 | 0.051 | 117462 | 0.88 | 0.57 | 0.94 | 1.25 |
| 8 | 14 | 140895 | 0.059 | 0.037 | 0.055 | 0.074 | 130539 | 1.11 | 0.83 | 1.14 | 1.39 |
| 9 | 16 | 156834 | 0.110 | 0.070 | 0.097 | 0.135 | 138741 | 1.17 | 0.95 | 1.19 | 1.43 |
| 10 | 16 | 130170 | 0.127 | 0.080 | 0.110 | 0.149 | 121626 | 1.12 | 0.84 | 1.14 | 1.44 |
| 11 | 16 | 99297 | 0.121 | 0.074 | 0.103 | 0.149 | 95661 | 1.10 | 0.83 | 1.11 | 1.38 |
| 12 | 16 | 83106 | 0.116 | 0.072 | 0.097 | 0.133 | 77235 | 1.03 | 0.72 | 1.05 | 1.36 |





| Longreach (23.44 S, BoM) | | | Aerosol Optical Depth, 500 nm | | | | Ångström exponent | | | | |
|---|---|---|---|---|---|---|---|---|---|---|---|
| Mon | Yrs | Nobs | Mean | 25Q | Median | 75Q | Nobs | Mean | 25Q | Median | 75Q |
| 1 | 3 | 20343 | 0.080 | 0.039 | 0.058 | 0.097 | 20343 | 0.75 | 0.45 | 0.83 | 1.10 |
| 2 | 3 | 21129 | 0.051 | 0.031 | 0.043 | 0.060 | 21129 | 0.75 | 0.46 | 0.72 | 1.02 |
| 3 | 3 | 24864 | 0.051 | 0.042 | 0.050 | 0.058 | 24864 | 0.70 | 0.51 | 0.74 | 0.91 |
| 4 | 3 | 32142 | 0.033 | 0.024 | 0.033 | 0.039 | 32142 | 0.94 | 0.60 | 0.88 | 1.18 |
| 5 | 3 | 29613 | 0.028 | 0.017 | 0.026 | 0.036 | 29613 | 0.71 | 0.38 | 0.66 | 0.95 |
| 6 | 3 | 23079 | 0.026 | 0.017 | 0.025 | 0.032 | 23079 | 0.34 | 0.13 | 0.59 | 1.00 |
| 7 | 2 | 22395 | 0.030 | 0.021 | 0.027 | 0.038 | 22395 | 0.38 | 0.04 | 0.43 | 0.75 |
| 8 | 2 | 27858 | 0.034 | 0.023 | 0.029 | 0.042 | 27858 | 0.51 | 0.14 | 0.47 | 0.88 |
| 9 | 2 | 24885 | 0.075 | 0.054 | 0.065 | 0.084 | 24885 | 0.58 | 0.30 | 0.69 | 1.29 |
| 10 | 2 | 28503 | 0.092 | 0.076 | 0.094 | 0.113 | 28503 | 0.82 | 0.44 | 0.97 | 1.30 |
| 11 | 2 | 18720 | 0.086 | 0.069 | 0.085 | 0.104 | 18720 | 0.49 | 0.23 | 0.51 | 0.82 |
| 12 | 3 | 22056 | 0.092 | 0.047 | 0.068 | 0.097 | 22056 | 0.78 | 0.57 | 0.84 | 1.16 |

| Alice Springs (23.80 S, BoM) | | | Aerosol Optical Depth, 500 nm | | | | Ångström exponent | | | | |
|---|---|---|---|---|---|---|---|---|---|---|---|
| Mon | Yrs | Nobs | Mean | 25Q | Median | 75Q | Nobs | Mean | 25Q | Median | 75Q |
| 1 | 15 | 164997 | 0.076 | 0.043 | 0.061 | 0.095 | 164997 | 0.61 | 0.27 | 0.54 | 0.90 |
| 2 | 15 | 147060 | 0.061 | 0.035 | 0.049 | 0.073 | 147060 | 0.63 | 0.35 | 0.59 | 0.87 |
| 3 | 15 | 153681 | 0.051 | 0.030 | 0.044 | 0.060 | 153681 | 0.64 | 0.36 | 0.57 | 0.82 |
| 4 | 15 | 169059 | 0.041 | 0.026 | 0.034 | 0.048 | 169059 | 0.85 | 0.59 | 0.82 | 1.08 |
| 5 | 15 | 161859 | 0.030 | 0.021 | 0.026 | 0.034 | 161859 | 0.98 | 0.71 | 0.92 | 1.14 |
| 6 | 15 | 164082 | 0.025 | 0.018 | 0.022 | 0.028 | 164082 | 0.83 | 0.61 | 0.82 | 1.04 |
| 7 | 15 | 200130 | 0.027 | 0.018 | 0.023 | 0.030 | 200130 | 0.94 | 0.73 | 0.93 | 1.15 |
| 8 | 15 | 227472 | 0.043 | 0.024 | 0.034 | 0.051 | 227472 | 1.04 | 0.83 | 1.05 | 1.25 |
| 9 | 15 | 197091 | 0.082 | 0.046 | 0.068 | 0.098 | 197091 | 1.16 | 0.93 | 1.17 | 1.38 |
| 10 | 15 | 193989 | 0.098 | 0.053 | 0.082 | 0.123 | 193989 | 1.08 | 0.84 | 1.13 | 1.34 |
| 11 | 15 | 153372 | 0.090 | 0.047 | 0.065 | 0.106 | 153372 | 0.96 | 0.67 | 0.94 | 1.23 |
| 12 | 15 | 131631 | 0.070 | 0.037 | 0.054 | 0.085 | 131631 | 0.70 | 0.48 | 0.70 | 0.92 |



| Birdsville (25.90 S, CSIRO) | | | Aerosol Optical Depth, 500 nm | | | | Ångström exponent | | | |
|---|---|---|---|---|---|---|---|---|---|---|
| Mon | Yrs | Nobs | Mean | 25Q | Median | 75Q | Nobs | Mean | 25Q | Median | 75Q |
| 01 | 9 | 27932 | 0.063 | 0.035 | 0.050 | 0.076 | 27932 | 0.59 | 0.22 | 0.56 | 0.92 |
| 02 | 9 | 22760 | 0.055 | 0.034 | 0.046 | 0.063 | 22760 | 0.61 | 0.27 | 0.61 | 0.93 |
| 03 | 10 | 34604 | 0.044 | 0.029 | 0.038 | 0.051 | 34604 | 0.50 | 0.25 | 0.50 | 0.74 |
| 04 | 9 | 30607 | 0.037 | 0.024 | 0.033 | 0.042 | 30607 | 0.78 | 0.51 | 0.78 | 1.04 |
| 05 | 9 | 26949 | 0.031 | 0.022 | 0.028 | 0.037 | 26949 | 0.98 | 0.69 | 0.97 | 1.21 |
| 06 | 9 | 23085 | 0.028 | 0.018 | 0.024 | 0.031 | 23085 | 1.01 | 0.69 | 0.98 | 1.27 |
| 07 | 10 | 26187 | 0.031 | 0.019 | 0.027 | 0.036 | 26187 | 1.06 | 0.78 | 1.07 | 1.33 |
| 08 | 10 | 33467 | 0.044 | 0.025 | 0.034 | 0.055 | 33467 | 1.21 | 0.93 | 1.20 | 1.45 |
| 09 | 11 | 40205 | 0.079 | 0.044 | 0.061 | 0.094 | 40205 | 1.23 | 0.95 | 1.25 | 1.53 |
| 10 | 10 | 38777 | 0.091 | 0.052 | 0.073 | 0.104 | 38777 | 1.13 | 0.80 | 1.20 | 1.50 |
| 11 | 9 | 24828 | 0.076 | 0.038 | 0.056 | 0.098 | 24828 | 0.97 | 0.61 | 1.03 | 1.36 |
| 12 | 9 | 26411 | 0.060 | 0.034 | 0.046 | 0.067 | 26411 | 0.63 | 0.26 | 0.62 | 0.96 |

| Geraldton (28.80 S, BoM) | | | Aerosol Optical Depth, 500 nm | | | | Ångström exponent | | | |
|---|---|---|---|---|---|---|---|---|---|---|
| Mon | Yrs | Nobs | Mean | 25Q | Median | 75Q | Nobs | Mean | 25Q | Median | 75Q |
| 1 | 9 | 118401 | 0.085 | 0.048 | 0.067 | 0.095 | 96831 | 0.40 | 0.12 | 0.39 | 0.67 |
| 2 | 9 | 96519 | 0.076 | 0.045 | 0.062 | 0.088 | 77730 | 0.47 | 0.17 | 0.42 | 0.73 |
| 3 | 9 | 89955 | 0.052 | 0.032 | 0.046 | 0.063 | 71049 | 0.49 | 0.19 | 0.46 | 0.79 |
| 4 | 9 | 72270 | 0.040 | 0.026 | 0.035 | 0.047 | 48900 | 0.68 | 0.34 | 0.71 | 1.03 |
| 5 | 10 | 69081 | 0.032 | 0.021 | 0.028 | 0.038 | 50034 | 0.62 | 0.18 | 0.63 | 1.06 |
| 6 | 10 | 77562 | 0.028 | 0.017 | 0.024 | 0.035 | 61611 | 0.47 | 0.11 | 0.53 | 0.86 |
| 7 | 10 | 75828 | 0.032 | 0.022 | 0.028 | 0.039 | 65628 | 0.51 | 0.09 | 0.52 | 0.94 |
| 8 | 9 | 75966 | 0.044 | 0.025 | 0.036 | 0.055 | 65964 | 0.69 | 0.28 | 0.71 | 1.12 |
| 9 | 9 | 73878 | 0.076 | 0.046 | 0.066 | 0.093 | 65076 | 0.87 | 0.59 | 0.95 | 1.22 |
| 10 | 9 | 92436 | 0.071 | 0.047 | 0.060 | 0.084 | 78741 | 0.80 | 0.50 | 0.84 | 1.14 |
| 11 | 11 | 97635 | 0.080 | 0.050 | 0.070 | 0.096 | 86379 | 0.62 | 0.34 | 0.63 | 0.92 |
| 12 | 11 | 136530 | 0.087 | 0.047 | 0.068 | 0.101 | 110145 | 0.64 | 0.18 | 0.47 | 0.82 |



| Tinga Tingana (28.98 S, CSIRO) | | Aerosol Optical Depth, 500 nm | | | | | Ångström exponent | | | |
|---|---|---|---|---|---|---|---|---|---|---|
| Mon | Yrs | Nobs | Mean | 25Q | Median | 75Q | Nobs | Mean | 25Q | Median | 75Q |
| 01 | 9 | 32799 | 0.086 | 0.044 | 0.064 | 0.093 | 32799 | 0.65 | 0.26 | 0.58 | 1.04 |
| 02 | 11 | 33001 | 0.059 | 0.032 | 0.045 | 0.070 | 33001 | 0.70 | 0.34 | 0.67 | 1.00 |
| 03 | 11 | 38212 | 0.049 | 0.027 | 0.039 | 0.056 | 38212 | 0.55 | 0.26 | 0.53 | 0.79 |
| 04 | 12 | 31238 | 0.038 | 0.024 | 0.032 | 0.043 | 31238 | 0.80 | 0.43 | 0.77 | 1.15 |
| 05 | 10 | 25510 | 0.038 | 0.020 | 0.025 | 0.034 | 25510 | 0.91 | 0.54 | 0.90 | 1.25 |
| 06 | 11 | 21402 | 0.026 | 0.018 | 0.023 | 0.029 | 21402 | 0.87 | 0.47 | 0.85 | 1.23 |
| 07 | 11 | 24220 | 0.027 | 0.018 | 0.024 | 0.032 | 24220 | 0.78 | 0.37 | 0.74 | 1.10 |
| 08 | 7 | 20779 | 0.043 | 0.025 | 0.034 | 0.051 | 20779 | 1.02 | 0.74 | 1.05 | 1.30 |
| 09 | 9 | 19911 | 0.093 | 0.043 | 0.070 | 0.114 | 19911 | 1.22 | 0.98 | 1.30 | 1.53 |
| 10 | 8 | 26343 | 0.073 | 0.036 | 0.050 | 0.078 | 26343 | 1.15 | 0.84 | 1.20 | 1.48 |
| 11 | 9 | 26199 | 0.063 | 0.033 | 0.043 | 0.063 | 26199 | 0.98 | 0.67 | 0.98 | 1.26 |
| 12 | 9 | 28923 | 0.064 | 0.033 | 0.046 | 0.065 | 28923 | 0.74 | 0.47 | 0.72 | 0.99 |

| Kalgoorlie (30.78 S, BoM) | | Aerosol Optical Depth, 500 nm | | | | | Ångström exponent | | | |
|---|---|---|---|---|---|---|---|---|---|---|
| Mon | Yrs | Nobs | Mean | 25Q | Median | 75Q | Nobs | Mean | 25Q | Median | 75Q |
| 1 | 7 | 78819 | 0.054 | 0.036 | 0.048 | 0.065 | 78819 | 0.92 | 0.54 | 0.80 | 1.23 |
| 2 | 7 | 66669 | 0.049 | 0.034 | 0.044 | 0.058 | 66669 | 0.80 | 0.52 | 0.76 | 1.05 |
| 3 | 7 | 71631 | 0.039 | 0.027 | 0.037 | 0.048 | 71631 | 0.74 | 0.44 | 0.70 | 0.99 |
| 4 | 7 | 47499 | 0.035 | 0.024 | 0.032 | 0.043 | 47499 | 0.85 | 0.54 | 0.83 | 1.18 |
| 5 | 7 | 45126 | 0.031 | 0.020 | 0.028 | 0.038 | 45126 | 0.81 | 0.48 | 0.88 | 1.17 |
| 6 | 7 | 41082 | 0.030 | 0.020 | 0.025 | 0.033 | 41082 | 0.94 | 0.52 | 0.98 | 1.30 |
| 7 | 6 | 44394 | 0.031 | 0.020 | 0.027 | 0.036 | 44394 | 1.01 | 0.71 | 1.02 | 1.33 |
| 8 | 6 | 46155 | 0.043 | 0.027 | 0.035 | 0.049 | 46155 | 1.07 | 0.82 | 1.12 | 1.37 |
| 9 | 6 | 52254 | 0.071 | 0.047 | 0.064 | 0.086 | 52254 | 1.19 | 0.91 | 1.27 | 1.51 |
| 10 | 6 | 72519 | 0.065 | 0.042 | 0.059 | 0.079 | 72519 | 1.18 | 0.87 | 1.19 | 1.48 |
| 11 | 6 | 60234 | 0.053 | 0.037 | 0.050 | 0.065 | 60234 | 0.98 | 0.77 | 1.01 | 1.21 |
| 12 | 7 | 93039 | 0.053 | 0.037 | 0.047 | 0.061 | 93039 | 0.80 | 0.50 | 0.75 | 1.08 |





| Fowlers Gap (31.09 S, CSIRO) | | Aerosol Optical Depth, 500 nm | | | | | Ångström exponent | | | | |
|---|---|---|---|---|---|---|---|---|---|---|---|
| Mon | Yrs | Nobs | Mean | 25Q | Median | 75Q | Nobs | Mean | 25Q | Median | 75Q |
| 01 | 3 | 10779 | 0.053 | 0.036 | 0.045 | 0.059 | 10779 | 0.95 | 0.70 | 0.96 | 1.18 |
| 02 | 3 | 10697 | 0.049 | 0.035 | 0.046 | 0.060 | 10697 | 0.83 | 0.59 | 0.83 | 1.09 |
| 03 | 3 | 9624 | 0.048 | 0.032 | 0.045 | 0.060 | 9624 | 0.84 | 0.64 | 0.81 | 1.02 |
| 04 | 3 | 6354 | 0.032 | 0.024 | 0.030 | 0.038 | 6354 | 0.93 | 0.67 | 0.92 | 1.16 |
| 05 | 3 | 5471 | 0.034 | 0.023 | 0.031 | 0.039 | 5471 | 0.74 | 0.38 | 0.76 | 1.06 |
| 06 | 3 | 5217 | 0.035 | 0.024 | 0.030 | 0.040 | 5217 | 1.00 | 0.78 | 1.06 | 1.27 |
| 07 | 3 | 6777 | 0.032 | 0.021 | 0.027 | 0.039 | 6777 | 1.01 | 0.62 | 1.09 | 1.33 |
| 08 | 3 | 10558 | 0.039 | 0.027 | 0.034 | 0.048 | 10558 | 1.18 | 0.94 | 1.19 | 1.45 |
| 09 | 3 | 10329 | 0.064 | 0.044 | 0.057 | 0.078 | 10329 | 1.19 | 0.99 | 1.22 | 1.41 |
| 10 | 2 | 7100 | 0.058 | 0.034 | 0.048 | 0.065 | 7100 | 1.12 | 0.87 | 1.14 | 1.41 |
| 11 | 2 | 6487 | 0.044 | 0.029 | 0.036 | 0.051 | 6487 | 0.99 | 0.82 | 1.00 | 1.15 |
| 12 | 3 | 8032 | 0.046 | 0.030 | 0.039 | 0.054 | 8032 | 0.86 | 0.69 | 0.84 | 1.01 |

| Woomera (31.16 S, BoM) | | Aerosol Optical Depth, 500 nm | | | | | Ångström exponent | | | | |
|---|---|---|---|---|---|---|---|---|---|---|---|
| Mon | Yrs | Nobs | Mean | 25Q | Median | 75Q | Nobs | Mean | 25Q | Median | 75Q |
| 1 | 3 | 39396 | 0.050 | 0.031 | 0.040 | 0.061 | 39396 | 0.93 | 0.59 | 0.94 | 1.33 |
| 2 | 4 | 42141 | 0.047 | 0.034 | 0.044 | 0.059 | 42141 | 0.92 | 0.64 | 0.90 | 1.17 |
| 3 | 4 | 43056 | 0.034 | 0.024 | 0.032 | 0.041 | 43056 | 0.87 | 0.49 | 0.89 | 1.21 |
| 4 | 4 | 28503 | 0.035 | 0.024 | 0.032 | 0.043 | 28503 | 0.86 | 0.61 | 0.88 | 1.13 |
| 5 | 4 | 25938 | 0.031 | 0.021 | 0.029 | 0.037 | 25938 | 0.81 | 0.43 | 0.78 | 1.19 |
| 6 | 4 | 22785 | 0.035 | 0.022 | 0.028 | 0.041 | 22785 | 0.70 | 0.36 | 0.75 | 1.09 |
| 7 | 3 | 22647 | 0.029 | 0.018 | 0.023 | 0.032 | 22647 | 0.91 | 0.59 | 0.96 | 1.19 |
| 8 | 3 | 27981 | 0.037 | 0.026 | 0.033 | 0.045 | 27981 | 0.83 | 0.52 | 0.90 | 1.14 |
| 9 | 3 | 34107 | 0.065 | 0.038 | 0.055 | 0.084 | 34107 | 0.95 | 0.67 | 0.98 | 1.25 |
| 10 | 3 | 39054 | 0.063 | 0.035 | 0.048 | 0.075 | 39054 | 0.76 | 0.44 | 0.82 | 1.17 |
| 11 | 3 | 35679 | 0.057 | 0.034 | 0.044 | 0.068 | 35679 | 0.96 | 0.68 | 0.96 | 1.22 |
| 12 | 3 | 38532 | 0.042 | 0.029 | 0.037 | 0.050 | 38532 | 0.77 | 0.50 | 0.80 | 1.04 |



| Lake Lefroy (31.25 S, CSIRO) | | | Aerosol Optical Depth, 500 nm | | | | Ångström exponent | | | | |
|---|---|---|---|---|---|---|---|---|---|---|---|
| Mon | Yrs | Nobs | Mean | 25Q | Median | 75Q | Nobs | Mean | 25Q | Median | 75Q |
| 01 | 4 | 9729 | 0.063 | 0.042 | 0.055 | 0.076 | 9729 | 0.75 | 0.50 | 0.68 | 0.97 |
| 02 | 4 | 9704 | 0.059 | 0.040 | 0.056 | 0.072 | 9704 | 0.78 | 0.53 | 0.78 | 0.98 |
| 03 | 4 | 8129 | 0.045 | 0.034 | 0.042 | 0.054 | 8129 | 0.79 | 0.57 | 0.77 | 0.99 |
| 04 | 4 | 6204 | 0.039 | 0.026 | 0.034 | 0.048 | 6204 | 0.93 | 0.69 | 0.95 | 1.18 |
| 05 | 4 | 6493 | 0.040 | 0.027 | 0.035 | 0.046 | 6493 | 0.95 | 0.62 | 0.97 | 1.25 |
| 06 | 4 | 7512 | 0.035 | 0.024 | 0.030 | 0.043 | 7512 | 1.24 | 1.04 | 1.28 | 1.50 |
| 07 | 4 | 7297 | 0.039 | 0.025 | 0.033 | 0.047 | 7297 | 1.02 | 0.64 | 1.11 | 1.37 |
| 08 | 4 | 6965 | 0.047 | 0.029 | 0.039 | 0.056 | 6965 | 1.03 | 0.75 | 1.04 | 1.31 |
| 09 | 4 | 9844 | 0.076 | 0.046 | 0.068 | 0.098 | 9844 | 1.12 | 0.87 | 1.15 | 1.44 |
| 10 | 3 | 10470 | 0.063 | 0.040 | 0.052 | 0.072 | 10470 | 1.06 | 0.85 | 1.08 | 1.31 |
| 11 | 3 | 5875 | 0.060 | 0.041 | 0.053 | 0.070 | 5875 | 0.91 | 0.70 | 0.91 | 1.13 |
| 12 | 4 | 9326 | 0.057 | 0.040 | 0.050 | 0.068 | 9326 | 0.73 | 0.49 | 0.72 | 0.94 |

| Cobar (31.54 S, BoM) | | | Aerosol Optical Depth, 500 nm | | | | Ångström exponent | | | | |
|---|---|---|---|---|---|---|---|---|---|---|---|
| Mon | Yrs | Nobs | Mean | 25Q | Median | 75Q | Nobs | Mean | 25Q | Median | 75Q |
| 1 | 3 | 35127 | 0.062 | 0.034 | 0.049 | 0.079 | 35127 | 0.79 | 0.49 | 0.76 | 1.03 |
| 2 | 3 | 23010 | 0.054 | 0.039 | 0.055 | 0.067 | 23010 | 0.75 | 0.57 | 0.79 | 0.97 |
| 3 | 3 | 27621 | 0.040 | 0.027 | 0.038 | 0.051 | 27621 | 0.68 | 0.33 | 0.65 | 1.04 |
| 4 | 3 | 20724 | 0.034 | 0.022 | 0.029 | 0.040 | 20724 | 0.78 | 0.44 | 0.74 | 1.07 |
| 5 | 3 | 19260 | 0.033 | 0.022 | 0.029 | 0.040 | 19260 | 0.64 | 0.41 | 0.66 | 0.88 |
| 6 | 4 | 15828 | 0.033 | 0.021 | 0.028 | 0.034 | 15828 | 0.82 | 0.55 | 0.86 | 1.14 |
| 7 | 3 | 22806 | 0.029 | 0.018 | 0.023 | 0.031 | 22806 | 0.62 | 0.38 | 0.58 | 0.84 |
| 8 | 3 | 27069 | 0.035 | 0.022 | 0.031 | 0.044 | 27069 | 0.67 | 0.37 | 0.65 | 1.01 |
| 9 | 3 | 33414 | 0.063 | 0.035 | 0.054 | 0.074 | 33414 | 0.44 | 0.03 | 0.68 | 1.17 |
| 10 | 3 | 35907 | 0.061 | 0.036 | 0.052 | 0.073 | 35907 | 0.90 | 0.55 | 0.96 | 1.28 |
| 11 | 3 | 31761 | 0.047 | 0.027 | 0.034 | 0.056 | 31761 | 0.74 | 0.51 | 0.74 | 0.98 |
| 12 | 3 | 33609 | 0.044 | 0.029 | 0.037 | 0.053 | 33609 | 0.72 | 0.43 | 0.68 | 0.94 |





| Mildura (34.24 S, BoM) | | | Aerosol Optical Depth, 500 nm | | | | Ångström exponent | | | |
|---|---|---|---|---|---|---|---|---|---|---|
| Mon | Yrs | Nobs | Mean | 25Q | Median | 75Q | Nobs | Mean | 25Q | Median | 75Q |
| 1 | 10 | 104439 | 0.059 | 0.033 | 0.044 | 0.068 | 104439 | 0.85 | 0.49 | 0.86 | 1.18 |
| 2 | 9 | 73968 | 0.061 | 0.035 | 0.052 | 0.075 | 73968 | 0.91 | 0.60 | 0.94 | 1.24 |
| 3 | 8 | 87930 | 0.047 | 0.033 | 0.042 | 0.057 | 87930 | 0.92 | 0.56 | 0.94 | 1.26 |
| 4 | 8 | 68325 | 0.044 | 0.032 | 0.041 | 0.052 | 68325 | 0.96 | 0.54 | 0.97 | 1.38 |
| 5 | 8 | 49818 | 0.044 | 0.029 | 0.040 | 0.052 | 49818 | 0.98 | 0.46 | 1.10 | 1.45 |
| 6 | 8 | 37413 | 0.038 | 0.024 | 0.030 | 0.041 | 37413 | 0.81 | 0.39 | 0.72 | 1.17 |
| 7 | 8 | 43722 | 0.040 | 0.024 | 0.034 | 0.046 | 43722 | 0.82 | 0.38 | 0.78 | 1.21 |
| 8 | 8 | 51516 | 0.049 | 0.032 | 0.043 | 0.056 | 51516 | 0.99 | 0.58 | 1.07 | 1.40 |
| 9 | 8 | 53826 | 0.081 | 0.053 | 0.073 | 0.097 | 53826 | 1.02 | 0.68 | 1.04 | 1.35 |
| 10 | 9 | 71685 | 0.064 | 0.044 | 0.054 | 0.074 | 71685 | 0.92 | 0.62 | 0.95 | 1.24 |
| 11 | 10 | 77835 | 0.052 | 0.035 | 0.045 | 0.059 | 77835 | 0.82 | 0.57 | 0.93 | 1.21 |
| 12 | 11 | 82419 | 0.052 | 0.033 | 0.042 | 0.055 | 82419 | 0.70 | 0.36 | 0.72 | 1.10 |

| Adelaide (34.95 S, BoM) | | | Aerosol Optical Depth, 500 nm | | | | Ångström exponent | | | |
|---|---|---|---|---|---|---|---|---|---|---|
| Mon | Yrs | Nobs | Mean | 25Q | Median | 75Q | Nobs | Mean | 25Q | Median | 75Q |
| 1 | 12 | 154815 | 0.060 | 0.038 | 0.053 | 0.073 | 154815 | 0.65 | 0.28 | 0.64 | 1.00 |
| 2 | 12 | 118848 | 0.058 | 0.038 | 0.050 | 0.067 | 118848 | 0.70 | 0.40 | 0.70 | 0.98 |
| 3 | 13 | 117720 | 0.048 | 0.033 | 0.045 | 0.059 | 117720 | 0.51 | 0.15 | 0.53 | 0.95 |
| 4 | 13 | 80235 | 0.049 | 0.032 | 0.044 | 0.059 | 80235 | 0.68 | 0.27 | 0.72 | 1.11 |
| 5 | 13 | 54048 | 0.048 | 0.032 | 0.042 | 0.058 | 54048 | 0.80 | 0.35 | 0.82 | 1.30 |
| 6 | 13 | 44610 | 0.047 | 0.029 | 0.040 | 0.055 | 44610 | 0.73 | 0.30 | 0.76 | 1.20 |
| 7 | 13 | 47583 | 0.046 | 0.028 | 0.041 | 0.056 | 47583 | 0.60 | 0.23 | 0.58 | 1.09 |
| 8 | 13 | 58410 | 0.059 | 0.038 | 0.052 | 0.072 | 58410 | 0.65 | 0.19 | 0.72 | 1.21 |
| 9 | 13 | 78237 | 0.080 | 0.051 | 0.071 | 0.097 | 78237 | 0.80 | 0.40 | 0.90 | 1.27 |
| 10 | 13 | 95628 | 0.075 | 0.050 | 0.066 | 0.091 | 95628 | 0.75 | 0.40 | 0.84 | 1.26 |
| 11 | 13 | 109044 | 0.064 | 0.042 | 0.055 | 0.075 | 109044 | 0.86 | 0.45 | 0.88 | 1.23 |
| 12 | 13 | 129108 | 0.063 | 0.038 | 0.052 | 0.078 | 129108 | 0.66 | 0.21 | 0.62 | 1.07 |



| Wagga (35.16 S, BoM) | | Aerosol Optical Depth, 500 nm | | | | | Ångström exponent | | | |
|---|---|---|---|---|---|---|---|---|---|---|---|
| Mon | Yrs | Nobs | Mean | 25Q | Median | 75Q | Nobs | Mean | 25Q | Median | 75Q |
| 1 | 15 | 148815 | 0.083 | 0.040 | 0.059 | 0.091 | 148815 | 1.03 | 0.64 | 1.06 | 1.45 |
| 2 | 16 | 114705 | 0.083 | 0.041 | 0.066 | 0.105 | 114705 | 1.14 | 0.83 | 1.18 | 1.51 |
| 3 | 16 | 144306 | 0.059 | 0.033 | 0.049 | 0.075 | 144306 | 1.07 | 0.73 | 1.10 | 1.45 |
| 4 | 16 | 114588 | 0.055 | 0.030 | 0.046 | 0.069 | 114588 | 1.26 | 0.95 | 1.32 | 1.61 |
| 5 | 16 | 90843 | 0.052 | 0.030 | 0.043 | 0.064 | 90843 | 1.22 | 0.88 | 1.32 | 1.61 |
| 6 | 17 | 58620 | 0.036 | 0.021 | 0.029 | 0.041 | 58620 | 1.15 | 0.76 | 1.19 | 1.56 |
| 7 | 16 | 60381 | 0.038 | 0.022 | 0.031 | 0.042 | 60381 | 1.11 | 0.74 | 1.15 | 1.54 |
| 8 | 17 | 86358 | 0.050 | 0.033 | 0.043 | 0.056 | 86358 | 1.08 | 0.73 | 1.16 | 1.46 |
| 9 | 17 | 109128 | 0.071 | 0.044 | 0.059 | 0.088 | 109128 | 1.06 | 0.76 | 1.13 | 1.43 |
| 10 | 16 | 126945 | 0.072 | 0.045 | 0.060 | 0.088 | 126945 | 1.04 | 0.72 | 1.07 | 1.39 |
| 11 | 17 | 115911 | 0.063 | 0.038 | 0.052 | 0.077 | 115911 | 1.11 | 0.79 | 1.12 | 1.40 |
| 12 | 17 | 154983 | 0.062 | 0.034 | 0.049 | 0.073 | 154983 | 0.95 | 0.60 | 1.01 | 1.34 |

| Canberra (35.27 S, CSIRO) | | Aerosol Optical Depth, 500 nm | | | | | Ångström exponent | | | |
|---|---|---|---|---|---|---|---|---|---|---|---|
| Mon | Yrs | Nobs | Mean | 25Q | Median | 75Q | Nobs | Mean | 25Q | Median | 75Q |
| 01 | 13 | 32164 | 0.096 | 0.053 | 0.077 | 0.110 | 32164 | 0.96 | 0.64 | 0.95 | 1.28 |
| 02 | 14 | 28148 | 0.090 | 0.049 | 0.074 | 0.105 | 28148 | 1.04 | 0.73 | 1.02 | 1.35 |
| 03 | 14 | 30960 | 0.067 | 0.036 | 0.056 | 0.089 | 30960 | 1.14 | 0.84 | 1.17 | 1.45 |
| 04 | 14 | 22939 | 0.071 | 0.034 | 0.051 | 0.089 | 22939 | 1.14 | 0.81 | 1.17 | 1.50 |
| 05 | 12 | 19717 | 0.046 | 0.027 | 0.036 | 0.050 | 19717 | 1.26 | 1.02 | 1.27 | 1.52 |
| 06 | 11 | 12527 | 0.038 | 0.024 | 0.032 | 0.044 | 12527 | 1.21 | 0.92 | 1.27 | 1.52 |
| 07 | 11 | 15065 | 0.039 | 0.024 | 0.032 | 0.044 | 15065 | 1.24 | 0.92 | 1.30 | 1.58 |
| 08 | 13 | 26419 | 0.058 | 0.032 | 0.045 | 0.064 | 26419 | 1.22 | 0.92 | 1.32 | 1.59 |
| 09 | 11 | 30892 | 0.071 | 0.043 | 0.061 | 0.086 | 30892 | 1.34 | 1.12 | 1.42 | 1.61 |
| 10 | 12 | 25773 | 0.086 | 0.050 | 0.073 | 0.104 | 25773 | 1.22 | 0.98 | 1.25 | 1.50 |
| 11 | 11 | 22616 | 0.070 | 0.042 | 0.061 | 0.086 | 22616 | 1.07 | 0.84 | 1.10 | 1.32 |
| 12 | 12 | 31646 | 0.068 | 0.041 | 0.057 | 0.084 | 31646 | 1.01 | 0.73 | 1.05 | 1.31 |