# Peer review of "The Climatology of Australian Aerosol"

_Atmospheric Chemistry and Physics, 2016_

## Referee Comment (RC1) · Anonymous Referee #3 · 3 Feb 2017

This is a valuable paper, and I am more than happy to recommend its publication in ACP. The data/climatology presented should be of interest to many scientists working in aerosols, and climate modelling. I have only a few small suggestions. Section 2 consists of a comparatively long "introduction", followed by a much shorter subsection 2.1. I feel the paper would be slightly more readable if that introduction were to be subdivided into 1 or 2 subsections, with appropriate headings to guide the reader. I found the concept of fitting both an annual cycle, plus two higher harmonics, quite interesting. However, an alternative way to characterize double-peaked data is to use two (or more) annual cycles with different phase angles. The reference list is comprehensive, however an additional reference the authors might like to consider is Bouya et al., Journal of Atmospheric and Solar-Terrestrial Physics, vol. 72, 726-739. I have one final thought for potential further work. The authors identify three aerosol 'groups': tropical, arid and temperate. Might it be possible to generate spatio-temporal patterns for each, perhaps via a "grand fit" to the data? (This would be somewhat similar to

[Figure]

EOFs.) This would then allow spatial interpolation of the climatology, although perhaps not extrapolation to Victoria and Tasmania.

---

## Referee Comment (RC2) · Anonymous Referee #1 · 11 Feb 2017

This work on "The climatology of Australian Aerosol" is very useful and covers a very critical gap, thus aerosol long timeseries for the southern hemisphere are very rare. Authors had made a great effort to collect and organise long timeseries of aerosol optical depth, and this work would be very useful for the scientific community. A very important part of this work is the discussion on the uncertainties raised from the use of different instruments. Analysis on these timeseries included the calculation of climatological values, the spectral decomposition and the classification of stations accordingly. All the above provide a robust scientific approach to the dataset that provides substantial results for the community. Methods and tools used are generally presented in detail and this approach could be applied in other data-sets of comparable time periods.

Climatological tables are of great usefulness, for future works and reference. This is a very important work, in the interests of ACP journal and it should seriously be considered for publication, after some minor improvements.

General comment: It is not clear how the stations is classified. It is only by figure 7 or it is primarily by the general climatology of each region. In 4.6 it is stated that "the grouping is based on figure 7". In that case, in the paragraph of 4.2 that investigates the unclassified station, could not have statements as " Tenant Creek is clearly tropical station". Even in case that geographical factors are considered, they should be indicated. Also, the discussion on 4.2 about the nature of each "unclassified" station, could not match up with the correlation found (table 7, discussion 4.7). Characteristics of each class is clearly stated only in the abstract section.

Section 2.1, line 20 page 6. "no statistically significant difference was found". Please explain in detail, which statistical significant approach was used and present some results on that.

Section 3. Before describing the tools used for the analysis, it is important to mention the quality control procedures applied on the timeseries. On section 2, the differences on cloud screening algorithms among the two types of instruments, are mentioned. Also selection on which months are used in the statistics is described in section 2. Were there any other filtering criteria applied on the values?

Section 3.1, line 15, page 7. Interpolated intervals should be smaller than the frequencies that is examined, otherwise significant noise is added to the signal. It is not cleared which tests were performed by authors, but the confidence that no significant noise was added should be clarified.

Section 4.2- Figure 7. It is not clear how threshold values for the classification was selected. It is not clear why Rockhampton station is out of the Arid box (other than its coastal location). Could it have a similar to Adelaide behaviour due to dust transport? The same for Tennant creek, which is very close to the tropical box. I think a method to objectify threshold values is needed in order to strengthen the classification. Maybe to have a hybrid criterion using also cycles information from section 4.3, could be a more objective approach.

Section 4.5. A useful addition could be a brief discussion on the four stations with statistical significant trends, and the nature of aerosols and emissions in their areas. Are all of them connected to AA emissions decline?

Figures 2-4. In cases of stations where statistical significant trends were recorded, these trends should be noted on timeseries plots also. Appendix, Climatological Tables: Angstrom exponent was calculated for different pairs of wavelengths at some stations. It should be noted on tables titles which pair was used on each station.

---

## Author Response (AR1)

**The Climatology of Australian Aerosol**

 **By Ross Mitchell, Bruce Forgan and Susan Campbell**

Manuscript acp-2016-1021, cover letter to the co-editor following recent reviews

Dear Co-editor,

We hereby resubmit the above manuscript following changes suggested by the two referees. In each case, our response to the referees is on the discussion web site, and is appended to this letter. With one exception, the changes made to the manuscript are given there.

The single exception concerns our response to referee #1, who raised the concern that "Characteristics of each class is clearly stated only in the abstract section". Our response was that "We accept the point that more detail on the characteristics of each class is given in the abstract than elsewhere, and have modified the manuscript accordingly".

In the final version of the manuscript as submitted, we have acted on this point by removing the detailed description of the class characteristics from the abstract, and providing a more descriptive account of the characteristics of the three classes in the Summary and Conclusions.

There are changes of a minor nature to the Acknowledgements section, and we have found and corrected several typographic errors.

We have uploaded the revised manuscript, together with a version highlighting the changes relative to the ACPD version, the latter called "diff.pdf". Note that the latter has been uploaded as a "supplement", file, as I do not know how to append a pdf file to a word document.

Let me know if further material is required.

Sincerely,

Ross M. Mitchell

17 March 2017

**The following contains concatenated reviews from anonymous reviewers #1 and #3, together with the authors' responses.**

**Reviews of "The climatology of Australian Aerosol" by R.M. Mitchell. B.W. Forgan and S.K. Campbell (**Manuscript acp-2016-1021)

**Response to anonymous referee #1**

The authors thank the referee for this constructive review, which begins by acknowledging the rarity of long time series of aerosol over the southern hemisphere, and the contribution made in presenting an aerosol climatology providing "substantial results for the community". From the outset, we emphasise that the primary goal of the paper is to present a continental-scale climatology of Australian aerosol. The classification work was intended to be exploratory in nature; unfortunately the referee has viewed this in a different light, and many detailed comments flow from this. As detailed below, we will amend the manuscript with this in mind.

 In order to address the points raised clearly, we have included the issues raised (in italics) before responding.

*General comment: It is not clear how the stations is classified. It is only by figure 7 or it is primarily by the general climatology of each region. In 4.6 it is stated that "the grouping is based on figure 7". In that case, in the paragraph of 4.2 that investigates the unclassified station, could not have statements as " Tenant Creek is clearly tropical station". Even in case that geographical factors are considered, they should be indicated. Also, the discussion on 4.2 about the nature of each "unclassified" station, could not match up with the correlation found (table 7, discussion 4.7). Characteristics of each class is clearly stated only in the abstract section.*

**Response**
As indicated by the title and in earlier remarks, this is primarily a paper that deals with the climatology of Australian aerosol, as determined from sun photometer measurements from the Bureau of Meteorology and CSIRO. The analysis of the time series via spectral decomposition supports the subsequent classification, which we regard as preliminary and exploratory. Undoubtedly, future studies will apply more objective clustering methods. Nevertheless, our approach is a necessary first step. In particular, the use of the relative AOD-Angstrom exponent amplitudes of the periodic component as a proposed classifier (Figure 7) is novel, and warrants closer consideration in subsequent work. The subsequent figures summarise the additional information contained in the relative amplitudes of the second and third harmonics (Figure 8) and the relative episodic to periodic variation caused by highly irregular smoke episodes (Figure 9). Both offer potential in future classification studies, although this is beyond the scope of this work. In the present work, these figures illustrate the role of the higher harmonics on the detailed shape of the spring-summer aerosol peak, and a means of separating the south-eastern stations based on the episodic to periodic aerosol component.

**Specific changes**: The sentence quoted in part by the referee will be changed to read:
"*Tennant Creek  resembles a tropical station with somewhat reduced Aτ, consistent with its location between Darwin and Alice Springs.*" No geographical factors were considered in the classification scheme. The discussion at the end of section 4.2 explains why the three stations Tennant Creek, Learmonth and Rockhampton are identified as "unclassified". The finding that several of the "unclassified" station pairs exhibit long-range correlation strengthens the case for continental-scale coherence in the time-variation of Australian aerosol. We accept the point that more detail on the characteristics of each class is given in the abstract than elsewhere, and have modified the manuscript accordingly.

*Section 2.1, line 20 page 6. "no statistically significant difference was found". Please explain in detail, which statistical significant approach was used and present some results on that.*

**Response**: The sentence identified by the referee has been changed to read *"The differences in the Angstrom exponents from the two different wavelength pairs were more than a factor of 4 less than the estimated standard uncertainties from the combined Type A and Type B uncertainties (ISO 1995) in the monthly mean Angstrom exponent*". [ISO: Guide to the expression of uncertainty in measurement, International Organization for Standardization, Switzerland, 101pp, 1995.]

*Section 3. Before describing the tools used for the analysis, it is important to mention the quality control procedures applied on the timeseries. On section 2, the differences on cloud screening algorithms among the two types of instruments, are mentioned. Also selection on which months are used in the statistics is described in section 2. Were there any other filtering criteria applied on the values?*

**Response**: The quality control methods are fully described in section 2. A more detailed comparison between the Bureau of Meteorology and CSIRO systems can be found in the cited reference Mitchell and Forgan (2003). No further filtering criteria were applied to the data sets, apart from those clearly stated in section 2.

*Section 3.1, line 15, page 7. Interpolated intervals should be smaller than the frequencies that is examined, otherwise significant noise is added to the signal. It is not cleared which tests were performed by authors, but the confidence that no significant noise was added should be clarified.*

**Response**: The interpolation algorithms used to fill in missing months in the time series were found not to introduce spurious frequencies into the FFT periodogram. This was determined by inspecting periodograms generated with three interpolation rules: linear, spline and Kalman. In no cases were spurious frequencies recorded. A stated in section 3.1, p.7, l.14, only spline and Kalman interpolation were used in the analysis presented in the paper. We stress that no interpolated data was used in deriving the fits to the time series using the model defined in equation (2).

*Section 4.2- Figure 7. It is not clear how threshold values for the classification was selected. It is not clear why Rockhampton station is out of the Arid box (other than its coastal location). Could it have a similar to Adelaide behaviour due to dust transport? The same for Tennant creek, which is very close to the tropical box. I think a method to objectify threshold values is needed in order to strengthen the classification. Maybe to have a hybrid criterion using also cycles information from section 4.3, could be a more objective approach.*

**Response**: As stated above, the main thrust of this paper is to define the climatology of Australian aerosol. As is clear from the above comment, there are many matters of detail left unresolved; these we are content to leave to future research. Nevertheless, the metrics defined here (detailed above in an earlier point) constitute an innovative approach to aerosol classification that will undoubtedly contribute to a more objective classification. We caution, however, that even "objective" clustering algorithms often require subjective decisions. For example, the "k-means" clustering method requires *a priori* input of the number of classes in the data, while in any method the "best" definition of the cost function requires subjective judgement.

**Specific change:** Line 12 of the abstract has been amended to read**: "***An exploratory classification of the aerosol types is undertaken based on (a) the relative periodic amplitudes of the Angstrom exponent and aerosol optical depth; (b) the relative amplitudes of the 6 and 4-month harmonic components of the aerosol optical depth, and (c) the ratio of episodic to periodic variation in aerosol optical depth."*

*Section 4.5. A useful addition could be a brief discussion on the four stations with statistical significant trends, and the nature of aerosols and emissions in their areas.*
*Are all of them connected to AA emissions decline?*

**Response**: We accept the referee's suggestion that some additional discussion is required regarding the stations showing trends, and propose to add the following.

**Specific change (text inserted following line 11 on page 11): "***The small negative trend at Alice Springs could well be due to the large aerosol peak in the austral summer 2002-2003, at the beginning of the Millennium drought, followed by fairly regular annual cycling thereafter (Figure 3). The more significant trend at Broome and Lake Argyle is considered given an expectation of increasing intensity of the monsoon in the north-west tropics (Rotstayn et al, 2009). Increasing monsoon rainfall can supress subsequent smoke emission due to lingering moisture in the vegetation,*

*although associated increased vegetation growth leads to increased smoke emission in subsequent dry seasons. Evidence for this is seen from Figure 2, in the suppressed aerosol immediately following the 2010-2011 monsoon (the largest seasonal rainfall on record), but with very large smoke emission during the following burning season (2012). The balance between these competing effects will determine the ultimate direction of any trend. The present analysis suggests that the trend may be negative."*

Note that the discussion around the RCP 4.5 decline in northern hemisphere (NH) AA emissions was introduced to give some context to the observed trends at the 4 stations. It simply demonstrates that the observed trends – possible reasons for which are briefly discussed (above) – are large by comparison with projected declines in NH AA emissions as experienced following transport into the southern hemisphere.

*Figures 2-4. In cases of stations where statistical significant trends were recorded, these trends should be noted on timeseries plots also. Appendix, Climatological Tables: Angstrom exponent was calculated for different pairs of wavelengths at some stations. It should be noted on tables titles which pair was used on each station.*

**Response**: Attention has been drawn to the two time series plots where significant trends were found, through the additional material added (p.11, see above point).
Climatological tables: The wavelength pairs used are given in the table caption. Further annotation cannot easily be given, as many of the climatological values use data both before and after 2009, when the change in wavelength pairs took place.

This concludes our response to referee #1. Our response to referee #3 follows.

This is a valuable paper, and I am more than happy to recommend its publication in
ACP. The data/climatology presented should be of interest to many scientists working
in aerosols, and climate modelling. I have only a few small suggestions. Section 2
consists of a comparatively long "introduction", followed by a much shorter subsection
2.1. I feel the paper would be slightly more readable if that introduction were to be
subdivided into 1 or 2 subsections, with appropriate headings to guide the reader. I
found the concept of fitting both an annual cycle, plus two higher harmonics, quite interesting.
However, an alternative way to characterize double-peaked data is to use
two (or more) annual cycles with different phase angles. The reference list is comprehensive,
however an additional reference the authors might like to consider is Bouya
et al., Journal of Atmospheric and Solar-Terrestrial Physics, vol. 72, 726-739. I have
one final thought for potential further work. The authors identify three aerosol 'groups':
tropical, arid and temperate. Might it be possible to generate spatio-temporal patterns
for each, perhaps via a "grand fit" to the data? (This would be somewhat similar to
EOFs.) This would then allow spatial interpolation of the climatology, although perhaps
not extrapolation to Victoria and Tasmania.

**Response**
The authors wish to thank the reviewer for these suggestions.

We agree that section 1 (Introduction) is long; however the material it contains is all required to set the work in context. Section 2 (Observations) contains a description of the different sun photometer and processing systems used by the two agencies concerned (The Bureau of Meteorology and CSIRO). This is necessarily long because two observing systems are described. However, it is difficult to shorten or subdivide it without compromising its coherence or content.

The reviewer acknowledges our representation of the aerosol time series using an annual cycle and two higher harmonics. As stated in the paper, this choice was based on the on the spectral decomposition achieved through a fast fourier transform (FFT). The reviewer suggests that the higher harmonics could be replaced by appropriately phased "annual cycles". If these annual cycles are sinusoids with periods of 12 months, then the time series cannot be adequately modelled since the details of the spring-summer aerosol peaks clearly require higher frequency variation than is available from 12-month sinusoids, however phased.  If the reviewer means pulsed or gated functions occurring every year, then the present analysis shows that this adds unnecessary complexity. On a monthly time scale, the time variation in aerosol can be well represented as a three-term harmonic oscillator. Since this finding has consequences for the construction of aerosol generation models, we will add a point to the Conclusions stating this.

We thank the reviewer for drawing out attention to the Bouya et al paper, and offering a suggestion for further work. We will consider the cited paper for inclusion. Coming up with a "grand-fit" to the data is a useful suggestion for further work, but well beyond the scope of the present paper.

Ross Mitchell, 10/2/2017

[revised manuscript text omitted]